

# 1 Are flood-driven turbidity currents hot-spots for priming
# 2 effect in lakes?

**D. Bouffard[1] and M-E. Perga[2,*]**
[1]{Physics of Aquatic Systems Laboratory, Margaretha Kamprad Chair, EPFL-ENAC-IEE-
APHYS, CH-1015 Lausanne, Switzerland}
[2]{INRA-Université Savoie Mont Blanc, UMR 042 CARRTEL, Thonon les Bains, France}
Correspondence to: M-E. Perga (marie-elodie.perga@thonon.inra.fr)
**Abstract**
In deep stratified lakes, such as Lake Geneva, flood-driven turbidity currents are thought to
contribute to the replenishment of deep oxygen by significant transport of river waters
supersaturated with oxygen into the hypolimnion. The overarching aim of this study was to
test directly this long-standing hypothesis. It combines direct observational data collected
during an extreme flooding event that occurred in May 2015 with dark bioassays designed to
evaluate the consequences of riverborne inputs on the hypolimnetic respiration. The
exceptional precipitations of May 2015 caused floods with annual return time for the Rhône
River, the dominant tributary of Lake Geneva, and with 50-year return time for the Dranse
River, the second most important tributary. Sediment loaded river flows generated turbidity
currents plunging into the lake hypolimnion. The observed river intrusions contributed to the
redistribution of dissolved oxygen, with no net gain, when occurring in the lowermost
hypolimnetic layer. In the uppermost hypolimnion above the last deep mixing event the
intrusions coincided with a net oxygen deficit. Consistent with field observations, dark
bioassays showed that 1% to 50% substitution of riverine organic matter to deep (<200 m)
hypolimnetic water did not affect microbial respiration, while addition of 1 to 10% of riverine
water to the uppermost hypolimnetic waters resulted in a respiration overyielding, i.e. excess
respiration of both riverborne and lacustrine organic matter. The results of our study conflict
the hypothesis that flood-driven turbidity currents necessarily increase hypolimnetic oxygen



stocks in Lake Geneva. In contrast, they show that flood-driven turbidity currents can be
potential hot-spots for priming effect in lakes.

## 1   Introduction

In thermally stratified lakes, river water inflow occurs under two different modes. Under
normal (i.e dry-weather) flow conditions, the river water is injected at the interface between
the warm, upper layer (epilimnion) and the cold lower layer (hypolimnion), forming an
interflow in the upper thermocline (Fischer, 1979) or at the surface. Particularly during flood
events, high concentrations of suspended sediments in rivers increase the density of inflowing
waters and therefore generate turbid density currents following the slope and flowing into the
hypolimnion (i.e. hyperpycnal plume). Their dynamics are divided into three distinct stages
(Alavian et al., 1992; Cortés et al., 2014; Hogg et al., 2013). First, the river dense water
pushes the ambient lake water, until the resulting baroclinic pressure created by the local
density difference between the river and the lake water balance the force of the momentum
inflow. At this stage, the river flow plunges (plunging stage), then the flow continues to run
along the lake bed as an underflow (underflow regime). The flow eventually reaches the depth
of neutral buoyancy, separates from the lake bottom and intrudes into the lake (intrusion
stage).
Lake observations of such riverborne turbidity currents date back to the late XIX[th] century by
Forel in Lake Geneva (Forel, 1892), and more recent reports indicate that they occur in many
perialpine lakes such as Walensee (Lambert et al., 1976), Lake Geneva (Lambert and
Giovanoli, 1988), Lake Lucerne (Wüest et al., 1988), Lake Brienz (Finger et al., 2006), and
Lake Lugano (De Cesare et al., 2006). Marine underflows are also common features (Mulder
et al., 2003).
Besides their implications on the physical structure of subaquatic environments (Meiburg and
Kneller, 2010), underflows have been considered for their potential biogeochemical
consequences on lakes. For instance, in Lake Geneva, the long-standing hypothesis has been
that river intrusions could replenish deep oxygen, and this hypothesis was supported by
several, albeit indirect, field observations. Fahrni and Rapin (1986) compiled the densities of
Lake Geneva hypolimnetic waters and those of the Rhône River over seven years and
suggested that some of the time-periods prone to river underflows coincided with partial
oxygen replenishment in the deepest layers of the lake's central area (long-term monitoring
station SHL2). Meybeck et al. (1991) pointed out relatively frequent and important oxygen





and silicate anomalies occurring close to the sediment-water interface at the reference
monitoring station SHL2 and invoked riverborne underflows along with alternative
mechanisms (accumulation of turbid, cold water on lake banks after severe storms) as being
responsible for these anomalies.
Lake Geneva has been suffering from deep water hypoxia since its early eutrophication in the
late 1950s (Jenny et al., 2014). Based on the hypothesis of oxygen-rich river water intrusions
(Meybeck et al., 1991), underflows following episodes of heavy rainfalls are expected to
counteract hypoxia by supplying oxygen to the stratified lake hypolimnion (Jenny et al,
2014). Yet, numerous hydroelectric dams have been constructed on the course of the Rhône
River, leading to a shift in the seasonal discharge pattern (Loizeau and Dominik, 2000).
Consequently, large floods mostly occurring in summer were reduced in amplitude and
frequency, also leading to a decrease in sediment input by at least a factor 2 (Loizeau and
Dominik, 2000). This summer discharge and bulk plume density reduction impacted the
occurrence of underflows along the lake bottom and is thought to aggravate deep water
hypoxia. However, despite their crucial functional implications on hypolimnetic oxygen
concentrations, the consequences of turbidity currents, and of their further decrease in
occurrence, have actually never been investigated directly.
Condition for the riverborne currents to reach the bottom of the central basin is that the
hyperpycnal plume does not get diluted by lake water as it advances within a lake (Turner,
1986). Yet, sediment loaded underflows, as those expected during floods, differ from
underflows observed during regular river regime, whose density is only controlled by
temperature. While the latter typically intrudes in the metalimnion as recently revealed by
isotope-tracing (Halder et al., 2013), the former undergoes a dynamic density change,
generating multiple underflows and therefore increasing the overall dilution (Cortés et al.,
2014). In addition, underflows might primarily act on the hypolimnion by pushing deep
waters upwards (Wüest et al., 1988), therefore redistributing oxygen within deep layers, rather
than contributing to a net oxygen gain, as observed in nearby Lake Lugano (De Cesare et al.,

28    2006).

Besides these positive and null hypotheses, an alternative, metabolic hypothesis is formulated
here based on the observed impact of floods on the carbon budgets of some lakes around the
world. A common opinion is that the bulk of the organic carbon that enters surface water is
refractory and may poorly contribute to bacterial metabolism, in comparison to autochthonous



sources (Moran and Hodson, 1990). However, the composition and hence quality of dissolved
organic matter entering lakes depend on the watershed coverage, land-use, climate and
hydrology (Alvarez-Cobelas et al., 2010) and might also be highly variable during the year
(Berggren et al., 2009). Increased inputs of fresh organic material during stormwaters and
spring floods (Agren et al., 2008; Dhillon and Inamdar, 2013; Raymond and Saiers, 2010)
have been shown to push lake metabolic balances further into heterotrophy (Klug et al., 2012;
Sadro et al., 2011; Tsai et al., 2008), either by decreasing surface primary production through
lower water transparency or by stimulating bacterial respiration through addition of labile,
terrestrial organic matter (Johengen et al., 2008; Ojala et al., 2011; Vachon and Giorgio,
2014). Considering the size of Lake Geneva, it is unlikely that floods may affect the whole
ecosystem metabolism (Vachon and Giorgio, 2014). Yet, since riverborne intrusions are rich
in organic matter in Lake Geneva, they could locally be hotspots for bacterial respiration that
would counteract the net oxygen inputs in the hypolimnion. In this metabolic hypothesis,
riverborne intrusions could cause a null or even a negative effect on hypolimnetic oxygen
concentrations.
The overarching aim of this paper was, therefore, to study the net oxygen effect of flood
driven riverborne intrusions in Lake Geneva. The study combined direct observational data
collected during an extreme flooding event in May 2015 while the lake was already stratified
supported with an experimental test designed to evaluate the consequences of river water
inputs on the hypolimnetic respiration.
**2    Materials and methods**
**2.1   Field survey**
Lake Geneva is the largest lake of Western Europe in terms of volume (89 km$^3$), and depth
(309 m). 84 % of the water input originates from the two main rivers, the Rhône (75 %) and
the Dranse (9 %), both flowing into the western basin. The river discharge and water quality
are continuously monitored by the FOEN (Federal Office for the Environment, Switzerland
for the Rhône), the DREAL (Direction Régionale de l'Environnement, de l'Aménagement et
du     Logement,     for     discharge     of     the     Dranse     River
http://www.hydro.eaufrance.fr/stations/V0334010) and the Observatory of Alpine Lakes,
respectively (for the water quality of the Dranse, http://www6.inra.fr/soere-ola).  Discharge
(hourly record) and water quality (two-weeks integrated sample) of the Rhône River are





monitored at Porte-du-Scex FOEN station 6 km upstream of the Rhône inflow. Discharge of
the Dranse is recorded at the Reyvroz hydrological station 20 km upstream of the Dranse inlet
to Lake Geneva while water quality surveys are performed close to the river delta.
While the Rhône river hydrological regime was originally of a typical glacier-type,
exploitation of the river flow for hydro-electrical production has substantially smoothed the
seasonal variability of water discharge over the latest 40 years, summer discharges being
currently only twice those observed in winter (Loizeau and Dominik, 2000). The average
annual discharge at Porte du Scex in May over 1935-2013 was 208 $m^3 s^{-1}$. The Dranse, which
is the second most important tributary of Lake Geneva, has a typical nival flow regime
(Meybeck et al., 1991), with maximum discharge in May (average discharge in May over
1906-2003: 39 $m^3 s^{-1}$).
The consequence of the heavy rain of May 2015 on physico-chemical lake properties was
investigated through a specific lake survey. 25 sites were sampled within one day with a
multi-parameter profiler (Sea&Sun Technology, CTD-90 multi-parameter probe), which
includes measurements of depth, temperature (T), conductivity, turbidity (Tu) and dissolved
oxygen concentration ($O_2$). The $O_2$ oxyguard (Clark type) is regularly calibrated with a long
term stability optode (Anderaa 4330F) but, in the present study, no drift in the $O_2$
measurements was expected during a single day survey.
The 25 sampling sites cover an area of ~200 $km^2$ over the Western basin. The sampling
design was specifically intended to sample the influence of the two main rivers and to
investigate the local influence of deep intrusions on $O_2$ concentrations. For this reason, all
sites were chosen with a local depth largely exceeding 100 m. Particular care was taken to
encompass the main active sub-lacustrine canyons of the Rhône and of the Dranse, as well as
their surroundings, and therefore to optimize the chance to probe intrusion plumes. The high
resolution CTD survey provided spatial information of $O_2$, Tu and T. Maps of hypolimnetic
properties were constructed with krigging interpolation method.
The net effect of river intrusion on the dissolved oxygen concentration was quantified by
comparing the $O_2$ profile within the intrusion layer to a theoretically-observed linearly
decreasing $O_2$ profile within this layer as typically observed in intrusion-free (undisturbed)
CTD profiles.



## 2.2 Oxygen consumption experiment

In order to test whether inputs of riverine organic carbon within the lake hypolimnion could drive significant oxygen consumption via microbial respiration, an incubation experiment was conducted in October 2015. On Oct 19[th], 15 L of Lake Geneva hypolimnetic water were collected with a VanDorn bottle from 100 m and 200 m depths at SHL2, such as 20 L of water from the Dranse, at less than 1 km from its entrance into Lake Geneva. One L of each lake and river water was kept in a glass bottle for further analyses of Carbon (C), Nitrogen (N) and Phosphorus (P) contents. Concentrations in total and dissolved organic C (TOC, DOC) were measured with/without filtration on Whatman GF/F filters (0.7 μm nominal pore size), on a TIC/TOC analyser (OI Analytical). Nutrients were analysed by standard colorimetric methods (Association Française de Normalisation, 1990).

Pure lake and riverine waters, and mixed waters in which different percentages of lake water was substituted by riverine waters, were incubated in 280 mL hermetically-closed glass-bottles equipped with SP-PSt3 planar oxygen-sensitive spots (PreSens), according to the experimental design presented in Table 1. Triplicates of each sample were incubated in a temperature-controlled dark chamber at 10 °C (a realistic temperature for river and hypolimnetic conditions during the flood). Change in $O_2$ over time were measured using a PreSens Fibox 3 equipped with a fiber optic oxygen transmitter. Initial $O_2$ concentrations were measured 1-hour after the start of the incubation, once water temperature had stabilized at 10 °C. Thereafter, $O_2$, and corresponding oxygen consumption, was measured once or twice per day for four days. Results were analysed by ANCOVA using $O_2$ consumption ($O_{2, 0}$ - $O_{2, t}$) as the response variable, treatment as the factor and time as covariate, including interactions. Further comparisons between treatments or specific dates were performed using Student's T test.

## 3   Results

### 3.1   Field survey

For the 2015 spring flood events, heavy rainfalls over the Lake Geneva watershed started on May 1[st] until May 3[rd] (total rainfall > 100 mm over these three days). For comparison, the City of Bex (Switzerland), located 20 km upstream of Lake Geneva along the Rhône River, collected 101 mm of rain over these three days, a record that had last been observed in December 1916. The discharge of the Rhône increased from ~140 $m^3 s^{-1}$ at the end of April up





to a maximum of 504 m$^3$ s$^{-1}$ on May 4$^{th}$ (Fig. 1a). This discharge reached the 98$^{th}$ percentile of
the cumulative distribution of the Rhône discharges of 1976 - 2009, and corresponded to an
annual return time of the Rhône floods at the entrance of Lake Geneva (Fig. 1b). During the
flood, the Rhône water temperature was 8 °C and O$_2$ concentrations were 11 mgO$_2$ L$^{-1}$ (data
source: FOEN, Switzerland). On the sampling date (May 7$^{th}$), the Rhône discharges remained
elevated with a daily average of ~ 400 m$^3$ s$^{-1}$.
The heavy rainfalls of early May 2015 increased the flow of the Dranse, which was already at
its seasonal maximum, concurring in generating a flood of exceptional amplitude. Between
April 30$^{st}$ and May 4$^{th}$, the discharge increased from 26 to 300 m$^3$ s$^{-1}$, right before the
monitoring station collapsed (Fig. 1a). This was a new record exceeding the previous
historical maximum discharge ever recorded at the Dranse hydrometric station, 229 m$^3$ s$^{-1}$ on
September 22$^{nd}$ 1968. The estimated return time of this 2015 flood event for the Dranse was
50 years (Fig. 1c). The Dranse waters during the flood were highly turbid with concentration
of suspended matter reaching > 2000 mg L$^{-1}$ (averaged concentration of 18 mg L$^{-1}$ in 2014;
data from the observatory of large lakes, France). The suspended organic matter concentration
was 195 mg L$^{-1}$ (annual average of 5 mg L$^{-1}$ in 2014). DOC was twice the average annual
concentration (2.3 vs 1.2 mgC L$^{-1}$). Dissolved nutrient concentrations were moderate (PO$_4^{3-}$ =
10 µgP L$^{-1}$; NO$_3^-$ = 480 µgN L$^{-1}$). Although not regularly monitored, the highly torrential flow
of the Dranse remained close to saturated oxygen concentrations.
The high turbidity of the inflowing flood waters compared with the background turbidity
signal of Lake Geneva (< 5 FTU), was thereafter used as a tracer for intrusive waters within
the lake. CTD profiles for all surveyed stations are provided as Appendix data (Figure A1).
Of the 25 sites, more than 50 % had hypolimnetic turbidity peaks attributed to river intrusions
(Fig. 2). The use of turbidity as a proxy for riverborne waters was also validated by
temperature profiles showing a consistent increase of temperature in the turbid layers and
thereby suggesting that locally the density of the water is significantly affected by susepnded
matters. This trend is clearly noticeable in BP18 located within the far Dranse underflow (Fig.
3a and b). The turbidity signal from the Rhône was restricted to the few stations located less
than 2 km downstream (BP8 and 9), while the turbidity current from the Dranse penetrated
much further within the lake, even reaching the reference monitoring station (SHL2 = BP18),
6 km downstream. The two northernmost stations BP2 and 3 were within the small but
noticeable underflow of the Veveyse River (Figure 2). Except for the stations closest to the



Rhône (BP8) and Dranse (BP21, 22 and 25), the underflow was split between two different
hypolimnetic layers: a very turbid underflow within the lower hypolimnion (below 110 m,
BP5, 7, 9 and 13 for the Rhône underflow, BP16 and 18 for the Dranse River), and less turbid
underflow between 50 and 110 m depths (BP2 and 29).
Undisturbed profiles typically indicated a similar trend in $O_2$ slope ($\Delta O_2/\Delta depth$ = -0.22 mg
$L^{-1}$ $m^{-1}$, SD 0.03 mg $L^{-1}$ $m^{-1}$, based on 9 profiles, BP13, 4, 6, 11, 12, 14, 19 and 28, Figure A1)
in the lower hypolimnion defined as the region below the winter deep mixing maximum (110
m) and the layer of influence of the bottom (20 m above the sediment interface). The winter
deep mixing maximum at depth 110 m was also confirmed by a CTD profile carried out one
week earlier by the Observatory of Alpine Lakes, France (not shown). Although more
contrasted, the upper hypolimnion, i.e. between the thermocline depth and the winter deep
mixing depth (Fig. 2b), was characterized by a less steep $O_2$ slope (e.g. the upper hypolimnion
mixed three months earlier).
The net effect of the intrusion on the $O_2$ was first assessed by comparing intrusion-affected
and nearest intrusion-free CTD profiles (Fig. 3). Surprisingly, at no sampling site did the
turbidity peak match with a local maximum in $O_2$ that could compare to the lens anomalies
reported by Meybeck et al. (1991). Instead, the depths of the turbidity peak coincided with a
disruption of the background decreasing trend in $O_2$ as a function of depth, clearly noticeable
for instance in the comparison of the $O_2$ profiles at BP18 (affected by the Dranse) and BP16
(not affected, Fig. 3a and b). Such a reduction of $O_2$ vertical gradients as recorded at BP5, 8, 9
and 18, suggests the formation of a mixed layer due to the increased momentum within the
underflow. Depth-averaged $O_2$ in the interval 30 to 270 m at BP16 and BP18 were identical
(7.0 $mgO_2$ $L^{-1}$) and, thereby, supported the hypothesis that the studied intrusion and related
extreme flood event had no net effect on the $O_2$ concentration but rather efficiently mixed
turbid-affected hypolimnetic layers. Although the trend was not as clear as in the middle of
the lake, similar conclusions arose from the comparison of CTD profiles carried out near the
Rhône at BP8 and BP12 (Fig. 3b and c). More information on the Rhône intrusion is provided
as Appendix data (Figure A2).
At stations BP2, 3, 21, 22, 25, and 29, turbid layers above 110-m depths even coincided with
a decrease in $O_2$ concentration. The drop in $O_2$ at BP2 in the turbid layer between 58 m and 86
m is a stunning example with a decrease in $O_2$ of ~0.3 $mgO_2$ $L^{-1}$. $O_2$ concentration decline



within turbid layers was also observed near the Dranse at BP21, 22 and 25 (Fig. 4). Although
all three stations were affected by the Dranse underflow, turbid intrusion was observed at
different depths. While $O_2$ of the three stations was highly comparable at depths unaffected by
the turbid flow (15 - 50 m), their $O_2$ profiles diverged at depths affected by the turbidity
current. Below 50 m, $O_2$ concentration at BP21 dropped as the turbidity increased, while $O_2$
concentration at BP22 and BP25 remained higher and similar between 50 m and 70 m. Below
70 m, $O_2$ concentration at BP22 dropped as turbidity increased and last, the turbidity intrusion
at 90 m depth in BP 25 coincided with the collapse of the three $O_2$ profiles (e.g. $O_2$ drop at
BP25). Surprisingly, below 110 m, $O_2$ profiles remains similar at the three stations
independently of turbidity values.
The difference in depth-averaged $O_2$ between measured profile and associated linear fit
through the turbid layer provided a first order parameterization of the net $O_2$ effect of the
intrusion. Note that due to the spatial heterogeneity in such large system, it was impossible to
define a single reference profile valid for the entire lake. Furthermore, the change in $O_2$ slope
at the winter deep mixing maximum (110 m) precluded the use of this linear fitting method
for any intrusion encompassing this layer (i.e. BP 21, 22 and 25) although they clearly
showed evidence for oxygen depletion within the turbid layer (see above). We therefore
restricted this analysis to intrusions located in the upper part of the hypolimnion (between the
thermocline and the winter deep mixing maximum) or intrusions located below this winter
deep mixing maximum. Relative changes in $O_2$ in the turbid layers flowing within the lowest
hypolimnion (> 110 m depth) were not significant (-0.07 g m$^{-3}$, SD 0.05 g m$^{-3}$, t = -2.50, df =
4, p-value = 0.066 ; Fig. 6). Net oxygen effects associated turbid layers flowing within the
upper hypolimnion were more variable (-0.19 g m$^{-3,}$ SD 0.16 g m$^{-3}$), but they were, on
average, significantly negative (t = -3.68, df = 7, p-value = 0.007), attesting of an actual
oxygen debt at these lower depths.
**3.2   Oxygen consumption experiment**
The experiment was designed *a posteriori* in order to explain observed differences in the
oxygen net effect of the Dranse intrusion between the upper and the lower hypolimnion
(above and below 110 m depth). In October 2015, DOC concentrations in the lake
hypolimnion and in the river were very similar (0.80 mgC L$^{-1}$ at 100 m depths, 0.70 mgC L$^{-1}$
at 200 m depths and 0.75 mgC L$^{-1}$ in the Dranse waters). Particulate organic carbon
concentrations were low (< 0.10 mgC L$^{-1}$). DOC in the Dranse waters during the experiment



were about three times lower than those observed during the flood but, more importantly,
DOC concentrations were highly comparable between dilution conditions. As a result,
differences in $O_2$ consumption between treatments cannot be driven by initial differences in
carbon contents.
Dissolved nutrient concentrations were very low in the Dranse at the time of collection.
Orthophosphate concentrations were half those recorded during the flood (5 µgP $L^{-1}$) while
nitrate concentrations were more similar (580 µgN $L^{-1}$). Orthophosphate concentrations at 100
m and 200 m depth were very comparable to those recorded during the flood (13 and 29 µgP
$L^{-1}$ respectively at both dates) while nitrate concentrations were slightly lower (620 and 560
µgN $L^{-1}$ in October, compared to 670 and 630 µgN $L^{-1}$, in May 2015).
Beyond 86 or 92 hours of incubations, some treatments (D100%, L200-100%, L200-99%)
presented a second phase of increased oxygen consumption that could indicate the start of
nitrification processes, i.e. oxygen consumption independent from aerobic mineralization. In
order to avoid any potential bias due to nitrification, final oxygen consumption values are
considered at 68 h of incubation. $O_2$ consumption over the first 68h was significantly different
between treatments (ANCOVA $F_{7,48} = 39$, $p < 2.10^{-16}$) and time ($F_{2,48} = 33$, $p = 8.10^{-10}$), with a
high consumption rate within the first 20 h, and a relative stabilization thereafter (Fig 6, a b).
$O_2$ consumption was the highest for the Dranse water, reaching 2.5 $mgO_2$ $L^{-1}$ after 68 h, while
final values of $O_2$ consumption were significantly lower for the lake waters, and within Lake
Geneva waters, $O_2$ consumption was higher at 200 m than 100 m depth (0.9 $mgO_2$ $L^{-1}$ and 0.5
$mgO_2$ $L^{-1}$ respectively, t = 4.0, *p = 0.02*).
Dilution of Lake Geneva water at 200 m depth with water from the Dranse (L200-100%,
L200-99%, L200-90% and L200-50%) did not significantly affect the dynamics of $O_2$
consumption over time (ANCOVA, $F_{treatment\ 3,84} = 2.0$, p = 0.10, Fig 6a. a). $O_2$ consumption
between treatments were therefore not significantly different after 15 h or beyond of
incubation of 200 m deep water. In contrast, $O_2$ consumption in lake water collected at 100 m
depths was higher for a treatment with 1-10% of Dranse water added, as compared to the non-
diluted samples (ANCOVA, $F_{treatment\ 2,66} = 96$, $p < 2.10^{-16}$). From 15 h of incubation and
beyond, $O_2$ consumption in samples incubated with 1-10% of Dranse water was significantly
(25 - 150%) higher than for undiluted samples, although initial carbon content was similar
between all treatments (Fig. 6b).



## 4    Discussion
### 4.1    River intrusions during the flooding event
Considering the extreme intensity of the observed rain event and subsequent river discharges
we expected the flood-induced turbidity current to be heavy enough to trigger an underflow
along the lake bed and therefore reach the deepest layers of the water column. However, no
clear signatures of a bottom following underflow could be observed for this specific event.
We estimated the sediment load in the Rhône river during the flood event by fitting the
relationship between river discharge, Q, and sediment load, C with a power law $C = aQ^b$ as
suggested in Loizeau and Dominik (2000) and Mulder et al. (2003). Our best fit for the 50
years of measurements resulted in $a = 5.7 \times 10^{-4}$ and $b = 2.36$ (see Appendix data, A3) which
is in good agreement with previously estimated relationships (Loizeau and Dominik, 2000).
Based on this relationship, the resulting estimated sediment load at the flood paroxysmal
phase reached 1.4 kg m$^{-3}$ (or g L$^{-1}$). Assuming that sediment load was predominantly made of
Quartz ($\rho_{sed} = 2700$ kg m$^{-3}$), the density of the Rhône river was estimated as $\rho_{R,tot} = \rho_w(S, T)$
$+(1-\rho_{sed} / \rho_w(S, T)) aQ^b = 1000.7$ kg m$^{-3}$, where $\rho_w(S, T)$ is the density of the water depending
on the temperature and salinity (Chen and Millero, 1986). This value was slightly lower than
the density of the lake water at the deepest location (1001.4 kg m$^{-3}$) and did not account for
the later entrainment of lake water into the intrusion. Similar estimates for the Dranse
provided $\rho_{R,tot} = 1001.2$ kg m$^{-3}$ assuming the same river temperature than for the Rhône.
Similar first order calculation suggests that the lower part of the intrusion stopped at ~160 m
for Rhône water and at ~250 m for Dranse, water which is in very good agreement with the
observations. Our results therefore confirm that Rhône discharge with annual return time is
actually plunging. However, the underflow may find its equilibrium density in the
hypolimnion and further evolve as an intrusion, rather than a true hyperpycnal current, for
which much higher discharges might be required. Recent observations of a strong turbidite on
Lake Geneva (Corella et al., 2014) were, for instance, interpreted as the result of a major
underflow and resulting landslides in October 2000 with an extremely strong Rhône discharge
of nearly 1400 m3 s$^{-1}$ (return time 300 years) and a sediment load of $> 9$ kg m$^{-3}$. Hence,
although Lambert and Giovanoli (1988) recorded 11 underflows associated with elevated or
rapidly changing discharge in the Rhône canyon, ~2.5 km away from the river mouth over a
short three months period in summer 1985, we could not time correlate any of these
underflows to any of the O$_2$ anomalies studied in the same year in the deepest 50 m of the





lake (269 - 309 m) by Meybeck et al (1991). This suggests that none of these underflows were
ultimately strong enough to travel far into the lake. Alternatively the previously postulated
relationship between $O_2$ anomalies and Rhône underflows as suggested by Meybeck et al.
(1991) is questionable as our observation shows that a one-year return time discharge rate
triggers a plunging underflow that will quickly degenerate into a deep intrusion. Underflows
evolving up to the centre of the lake require strong discharge with return times longer than
one year and are therefore infrequent. Due to its closer location to the lake centre, the Dranse
is more likely to affect the lake centre (BP18, SHL2) with interflow (present study) or
underflow (Meybeck et al. 1991).
**4.2   Consequences of river intrusions on hypolimnetic oxygen concentrations**
Overall, the dataset presented herein rejected the hypothesis of a net oxygen gain due the river
intrusions in Lake Geneva during this important flooding event.
When comparing our results to those that supported this original hypothesis, it seems that the
net effect of river intrusions on hypolimnetic oxygen concentrations of Lake Geneva varies
depending on the properties of the intrusion flow. In May 2015, neither the Dranse nor the
Rhône generated an underflow plunging to the lake bottom and we cannot exclude that such
hyperpycnal flows could indeed contribute to deep water oxygen replenishment. Instead, we
observed two types of intrusions, i.e. in the upper hypolimnion that had been previously
mixed during the preceding winter, and in the lower hypolimnion.  Intrusions in the lower
hypolimnion acted essentially through their mixing momentum and partly redistributed
oxygen within the mixed layers with, yet, no net benefit. Intrusions above 110 m depths
consistently generated local oxygen depletion. Before the flood, $O_2$ concentrations in the
upper hypolimnion were higher than in the lower hypolimnion (9.5 $mgO_2$ $L^{-1}$ and < 7 $mgO_2$ $L^{-}$
$^1$ respectively) because the winter mixing did not reach deeper than 110 m that year. The $O_2$
concentrations in the Rhône were as high as 11 mg $L^{-1}$ and we assumed that the Dranse waters
were also slightly supersaturated. It is then unlikely that the observed oxygen depletion in the
turbidity current observed in the upper hypolimnion directly resulted from intrusions of $O_2$-
depleted river waters. Instead, they point to increased oxygen consumption in the uppermost
turbidity current, for which the metabolic consequences of the riverborne inputs would have
taken over its physical, mixing effect.





These observations suggested (i) that respiration of diluted, riverborne organic matter in the
hypolimnion had a significant effect on oxygen concentrations and (ii) that the contribution of
respiration varied between the upper and lowermost hypolimnion. The oxygen consumption
experiment that we designed, *a posteriori*, aimed at testing whether such assumptions were
reasonable. Although $O_2$ and DOC concentrations in the hypolimnetic waters were likely to
be relatively similar in October to those right before the flood, they were undoubtedly quite
different for the river waters. However, this experiment did not intend to mimic conditions
during the flood but instead to investigate the variability of the metabolic processes in the
different hypolimnetic layers.
**4.3  Hypolimnetic respiration of riverborne organic matter**
Microbial respiration for the Dranse water was initially three times those of the lake
hypolimnetic waters, for similar DOC concentrations. In order to best reproduce processes
occurring during the river intrusion in the lake, we did not filter water to remove plankton
before incubations, in contrast to experiments aiming at separating bacterial and planktonic
respiration rates (Warkentin et al., 2007). It is therefore likely that the respiration was higher
in the Dranse water samples because it included both bacterial and autotrophic planktonic
components compared to samples from the dark hypolimnetic layers in which the sole
microbial heterotrophs shall be present.
However, considering a respiratory quotient of 0.82 (Williams and del Giorgio, 2005),
consumed $O_2$ in the Dranse river samples after 68 h (i.e. 78 $\mu molO_2\ L^{-1}$, i.e. 1.14 $\mu molO_2\ L^{-1}$
$h^{-1}$) would correspond to the oxidation of 0.75 $mgC\ L^{-1}$, i.e. > 90 % of TOC. These values of
short –term oxygen consumption rates belong to the upper end of the range reported for lakes
and streams by Berggren et al. (2012). They attest of an important short-term labile pool of
DOC (*sensu* Guillemette and del Giorgio (2011)) in the river waters (low-molecular weight,
relatively young DOC, (Agren et al., 2008)) but also of low bacterial growth efficiency due to
nutrient limitation in the oligotrophic conditions of the Dranse rivers (Cimbleris and Kalff,
1998; Wiegner and Seitzinger, 2004). River water samples were collected for the experiment
purposes at times of moderate hydrological loads and DOC as well as phosphate
concentrations in the river during the flood were much higher suggesting fast leaching of the
watershed soils (Agren et al., 2008). It is likely that river DOC during the flood was even
more labile, since it was mobilized and transported by rapid flush and fast transport of soil
organic matter (Agren et al., 2008; Bergström and Jansson, 2000).



In contrast, respiration recorded in the lake hypolimnetic waters corresponded with lower
oxidation rates (23 % and 42 % of TOC at 100 m and 200 m depths, respectively) for similar
initial organic carbon contents. Bacterial growth (including respiration, production, and
growth efficiencies) depends both on nutrient limitation and organic matter quality (Farjalla et
al., 2009). At these depths, microbial metabolism is less likely to be nutrient limited but
bacterial abundances are nevertheless usually low (around $10^5$ cells ml$^{-1}$, S. Jacquet, pers.
comm.) suggesting low values for bacterial production. Besides, hypolimnetic waters of Lake
Geneva have long-residence times (time of ~20 years (Meybeck, 1970) and although most of
the lakes' hypolimnetic DOC might primarily originate from autochthonous primary
production, DOC aging through microbial reworking contributes to increasing its aromaticity
(Berggren et al., 2009) resulting in low bacterial growth efficiencies even without nutrient
limitation (Berggren et al., 2009). In that case, lower respiration values for hypolimnetic
waters suggested that lake DOC was semi-labile as compared to the Dranse DOC that might
be fresher and more readily available.
Nevertheless, higher oxygen consumption rates measured for the lower hypolimnion as
compared to its upper layer are surprising at first sight as they point to a higher availability for
lake DOC of greater depths. Such depths–related differences in C availability for microbial
metabolism are also consistent with the substitution assays showing that riverborne, labile
DOC inputs stimulated microbial respiration only for the supposingly C-limited samples, i.e.
the 100m depth lake water (Eiler et al., 2003). If DOC had the same sources in both
hypolimnetic layers, the greater water retention time would instead contribute in decreasing
DOC bioavailability with depth. Yet, DOC concentrations increases between the lower limit
of the mixed hypolimnion (110 m) to the lake bottom (309 m), from 0.7 to 0.8 mgC L$^{-1}$, as a
likely consequence of DOC remobilization from the sediment and accumulation in the
overlaying water column (Gonsior et al., 2013). Recent studies highlighted DOC release from
the sediment is a substantial source of labile DOC to the water column (Downing et al.,
2008), which could increase to the short-term labile pool of DOC in the unmixed
hypolimnion. While additional investigation on deep DOC quality would be required, fluxes
of sediment DOC to the unmixed deeper hypolimnetic layer could sustain higher respiration
rates as compared to the most superficial one for which microbial metabolism is the most
limited by organic matter quality.





**4.4  Excess respiration in the mixed samples and overyielding**
More surprisingly though, the stimulation of microbial respiration for the 100-m depth
treatment was disproportionate as compared to the quantity of added labile OC. The
substitution of 1 % of lake DOC by riverborne, more labile DOC almost doubled the
respiration rate within 68 hours. Substituting 1 % of the lake water from 100 m depth with
Dranse water is predicted to generate an excess $O_2$ consumption of 0.018 $mgO_2$ $L^{-1}$ based on
the respiratory values of the D-100% samples. However, the experimentally observed values
were +0.5 $mgO_2$ $L^{-1}$, showing a clear overyield (*sensu* Farjalla et al, 2009). Similarly,
substitution of 10 % of lake DOC increased oxygen consumption by 1 $mgO_2$ $L^{-1}$ while
proportionality suggested instead the value of 0.18 $mgO_2$ $L^{-1}$.
Such effects were documented by Farjalla et al. (2009) who observed that a mixture of fresh
and aged DOC acted synergistically on the bacterial respiration rate (Farjalla et al., 2009)
resulting in disproportionately increased rates compared to single substrates. We did not filter
the Dranse water to remove microbes prior to incubations, with the purpose of more closely
replicating realistic conditions within turbidity currents, and it is likely that we added an
inoculum of river microbes to the mixture experiments. A greater microbial diversity in the
mixture samples could favour co-metabolism on carbon compound decomposition and
therefore the observed overyielding, as suggested by Farjalla et al. (2009). Yet, more recent
investigations revealed that the initial microbial community composition has less impact of
DOC use than the nature of DOC itself (Attermeyer et al., 2014). The microbial riverine
inoculum might then not account for the totality of the enhanced decomposition of DOC in
the mixture, while metabolic synergies in the microbial use of the different DOC qualities
could also be involved (Fonte et al., 2013).
**4.5  Experimental and observational conclusions on the effect of river**
**intrusion on the hypolimnetic oxygen concentrations of Lake Geneva.**
The May 2015 flood episodes did not trigger a true underflow. Therefore, we cannot generally
exclude that underflows, in which very high turbidity limits the mixing of the water masses of
the river and the lake, can finally replenish deep water oxygen. However, rough estimations
confirmed that such truly underflow processes are far more rare than previously thought.
Exceptional events that indeed replenish oxygen at the bottom of the lake might occur at
decennial, rather than annual time scales. Our observations pointed to null or negative effect



of river intrusions on the deep water oxygen content of Lake Geneva. Rather than increasing
deep water oxygen concentrations these intrusions cause physical mixing of the deep
hypolimnion, i.e. redistributing oxygen over depth, or have a metabolic effect. The final
consequences on the benthic biota are unclear since the effect on $O_2$ might be transient and
shall not persist for long within the lake hypolimnion. As a matter of fact, the change in the
deep hypolimnion $O_2$ profiles due to the homogenization by the intrusion was poorly visible
during the monitoring survey performed at SHL2 a few days after the end of the flooding
event (11 May, data not shown). While the underlying mechanisms explained why the relative
contribution of physical mixing and metabolism varied with the intrusion depth, our
observational survey and bioassays highlighted that these intrusions provide interfaces where
riverine and lake organic matter are mixed and can act as biogeochemical hotspots. Since the
quantity of substituted DOC does not account for the excess oxygen consumption observed in
the L100-99% and L100-90% treatments, it means that more of the 100 m depth,
hypolimnetic DOC had been respired as a small fraction of more labile, riverborne, DOM was
substituted. Even if the role played by a potential inoculum of riverine microbes cannot be
ruled out, river intrusions in the upper hypolimnion resulted in an increase of both
autochthonous and allochthonous organic matter respiration. This mechanism by which a
small addition of labile organic matter stimulates the mineralization of less available organic
matter is referred as to 'priming effect'. Priming effect has been thoroughly investigated in
soils but evidence are still sought in aquatic systems (Bianchi, 2011; Catalán et al., 2015).
River plumes had been identified as sites prone to host priming effect, since they shall bring
in contact different sources of organic matters with varying quality (Bianchi, 2011; Guenet et
al., 2010). Even though several papers, along with the present results, have revised the long-
standing hypothesis of the recalcitrance of terrestrial organic matter (Guillemette and del
Giorgio, 2011; Roehm et al., 2009), investigations of aquatic priming effect are still based on
such hypothesis that lacustrine OM shall be the primer and terrestrial OM the "primed"
(Catalán et al., 2015).  In the present study, the vertical consideration of the intrusion
challenged our preconceived thoughts on the quality of aquatic DOC, showing that lacustrine
DOC was semi-labile, but of heterogeneous quality with depths, while river OM was
potentially acting as the primer. Overall, deep river intrusion in Lake Geneva might therefore
be potential hotspots for aquatic priming, fostering the mineralization of deep, less labile
lacustrine organic matter.





**Appendix**
A1. CTD profiles
A2. CTD transect from the Rhône mouth to the lake centre. These series of CTD profiles
suggest that the net $O_2$ effect of the Rhône intrusion is limited or null but instead efficiently
homogenized the $O_2$ in the hypolimnetic water affected by the intrusion (reduction of the $O_2$
gradient close to the Rhône River). Intrusion are associated with elevated Tu signal.
A3. Sediment rating curve of the Rhône River at La Porte-du-Scex from 1964 to 2015. Best
fit Concentration [mg L$^{-1}$] = aQ$^b$, with Q = discharge [m$^3$ s$^{-1}$], yields a = 5.7 x 10$^{-4}$ (3.1 x 10$^{-4}$,
8.3 x 10$^{-4}$) and b = 2.365 (2.29, 2.441) with the 95% confidence interval in parenthesis.
A4. GPS coordinate (geodetic datum CH1903+) of the CTD profiles carried out on May 7$^{th}$

16 2015

**Author contribution**
Both authors contributed equally to the field work, the data analysis and the redaction of the
manuscript.
**Acknowledgements**
The authors thank Johny Wüest, Martin Schmid and Beat Müller for their comments on a
previous draft. The authors also thank Robert Schwefel for his help in the field.



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



**Figures captions**
Fig. 1. Hydrological characteristics of the flooding event of May 2015. a. Daily precipitation
and discharges of the two dominant tributaries of Lake Geneva, the Rhône and the Dranse
rivers over the months preceding and following the flood. The break in the discharge record
of the Dranse corresponds to the date at which the station collapsed. Cumulative distribution
of the Rhône discharges in 1976-2009 (b) and Dranse discharges 1957-2014 (c). Red lines
indicates the maximum discharges of the May 2015 event.
Fig. 2a. Spatial distribution of hypolimnetic turbidity (40 - 300 m depth) as a tracer for
flooding river intrusions. Identified stations are those for which profiles were provided in
Figure 3b. The inserted figure shows typical temperature and $O_2$ profiles for an undisturbed
station. Note the linear decrease of $O_2$ with depth in the lower hypolimnion (below the deep
winter maximum). GPS location of the CTD profiles is indicated in the Appendix information
(A4).
Fig. 3. Comparison of temperature, turbidity and $O_2$ depth-profiles for nearby stations, one
being undisturbed (dotted lines), and the other highly disturbed (continuous lines) by the
turbidity current of the Dranse (a,b); the Veveyse (c,d) and the Rhône rivers (d,e,).
Fig. 4. Comparison of turbidity (a) and $O_2$ (b) depth-profiles for the three stations close to the
Dranse river mouth. The shaded area corresponds to the upper hypolimnion, i.e. water layers
that have been mixed during the preceding winter.
Fig. 5. Net effect of the turbidity layer on $O_2$ concentrations calculated for intrusion above
and below the deep winter winter maximum (110 m).
Fig. 6. $O_2$ consumption in the bioassays. a. Bioassays conducted for the lake water collected in
the lowermost hypolimnion at 200m depth (100% 200 m), and with 1, 10% and 50%





substitution, respectively, with Dranse water (99% 200 m, 90% 200 m and 50% 200 m,
respectively, and compared for Dranse water only (D 100%). b. Bioassays conducted for the
lake water collected in the uppermost hypolimnion at 100 m depth (100% 100 m), with 1 and
10% substitution, respectively, with Dranse water (99% 100 m, 90% 100 m respectively), and
compared for the Dranse water (D 100%).



**Table 1. Design of the incubation experiment.**

| Sample designation | Percent composition Lake Water | | |
|---|---|---|---|
| | 200m-depth lake water | 100m-depth lake water | Dranse water |
| L200-100% | 100% | | |
| L200-99% | 99% | | 1% |
| L200-90% | 90% | | 10% |
| L200-50% | 50% | | 50% |
| L100-100% | | 100% | |
| L100-99% | | 99% | 1% |
| L100-90% | | 90% | 10% |
| D-100% | | | 100% |





3    **Figures**

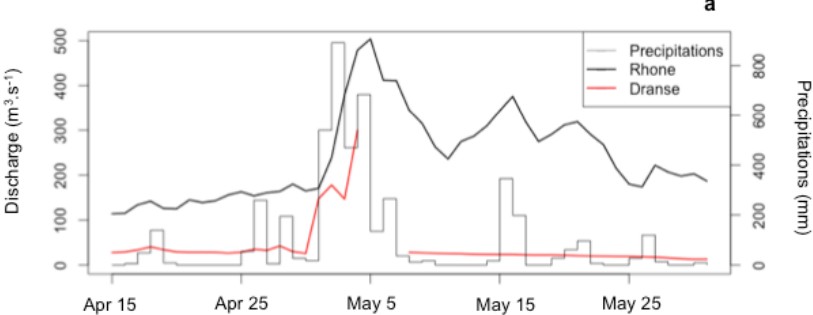

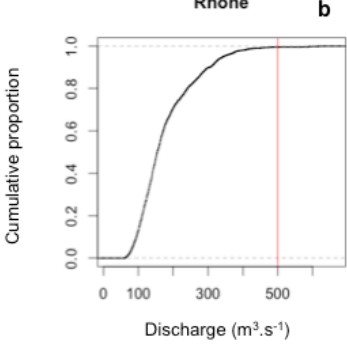 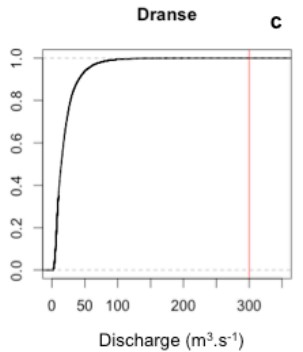

6    Figure 1



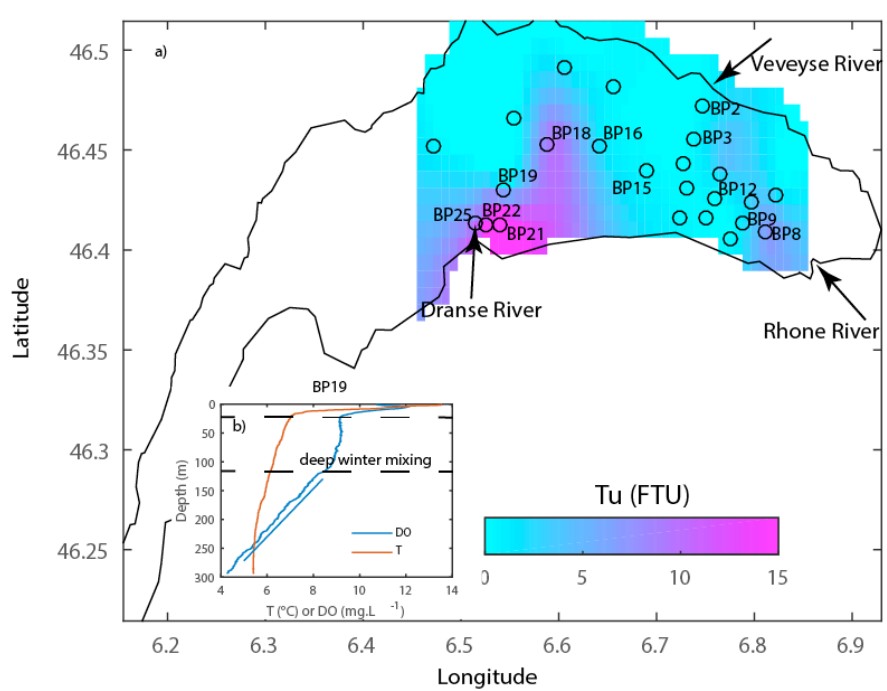

2      Figure 2



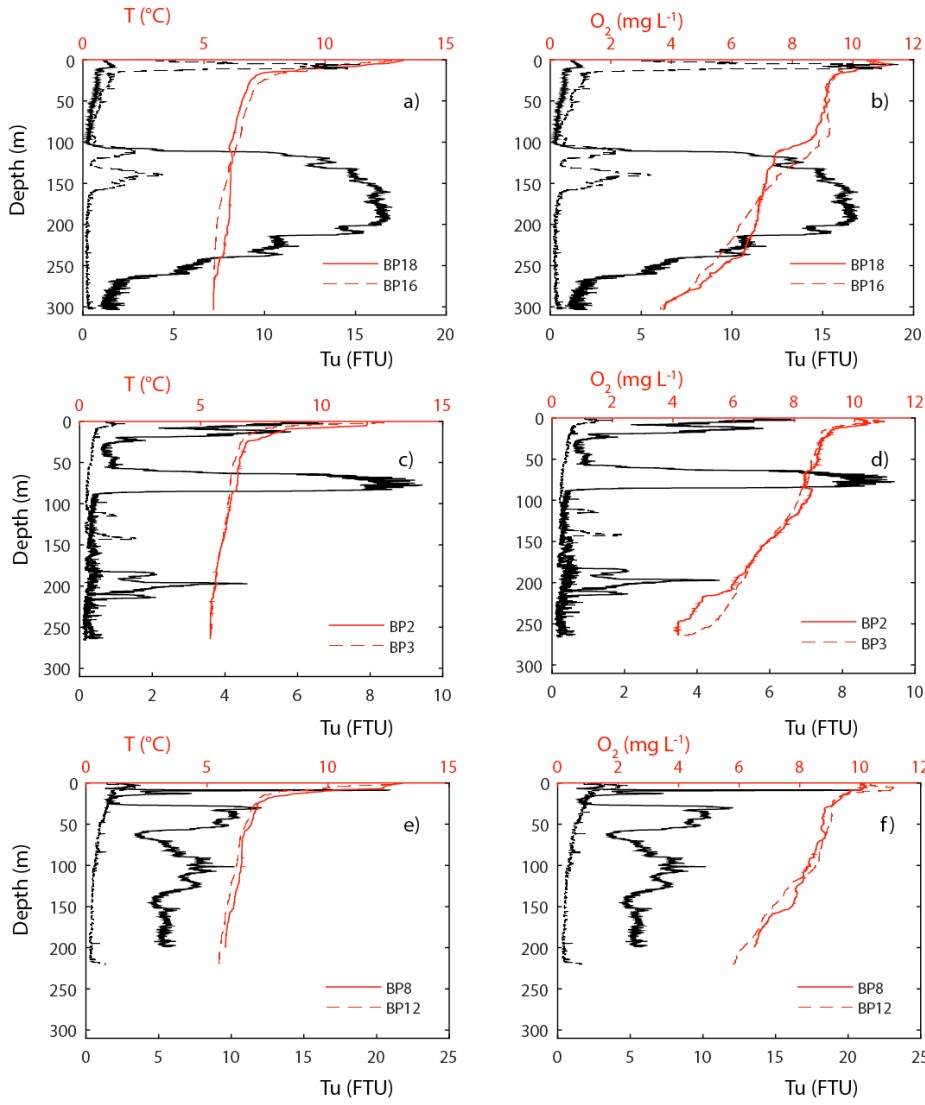

4    Figure 3





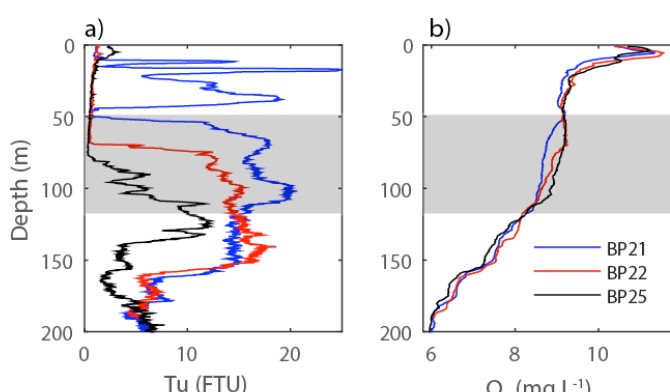

4    Figure 4

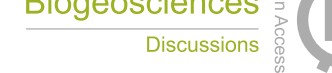

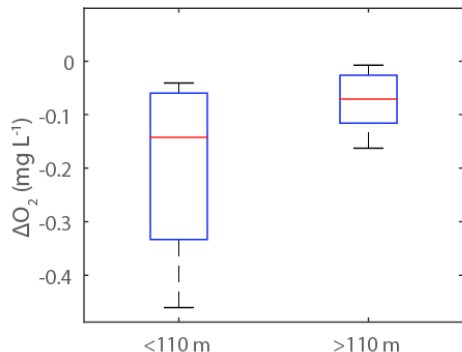

5    Figure 5



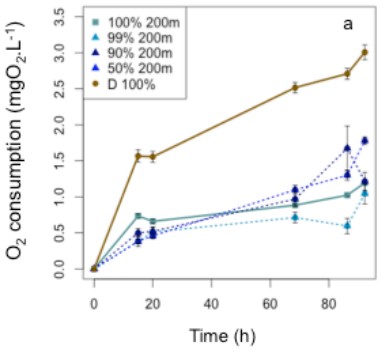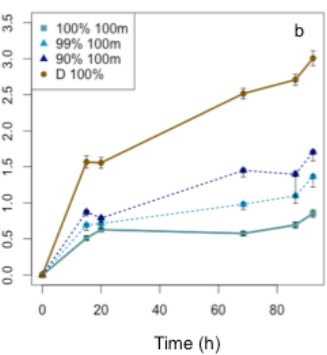

3    Figure 6





1 **Appendix**

2 A1.

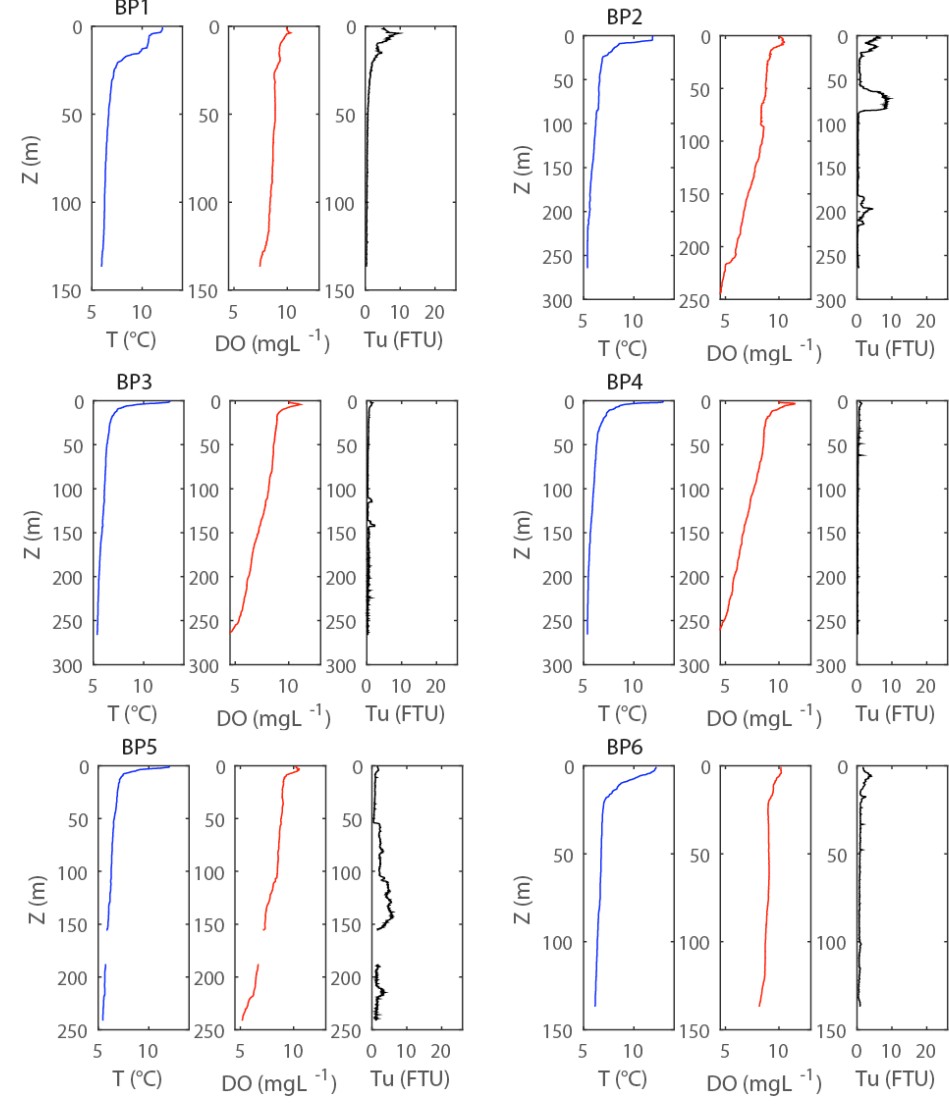





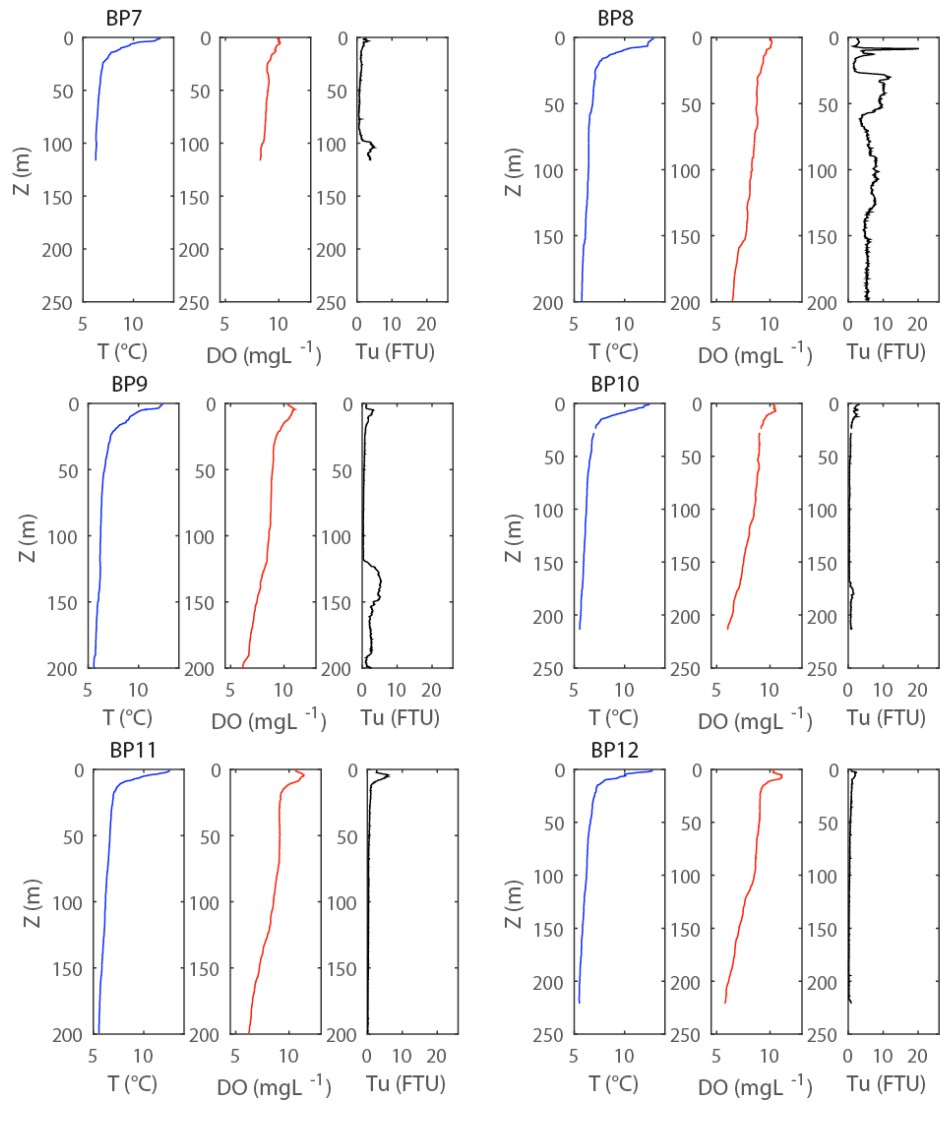



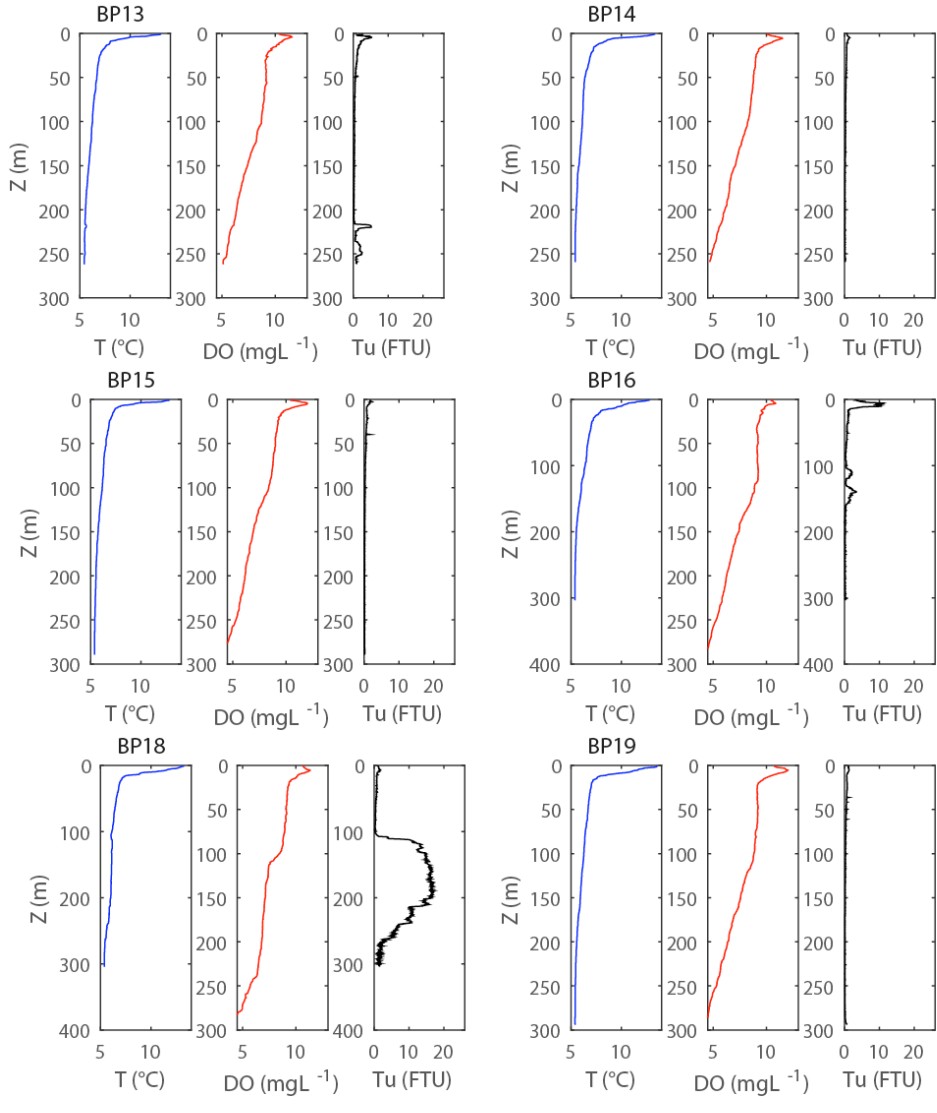





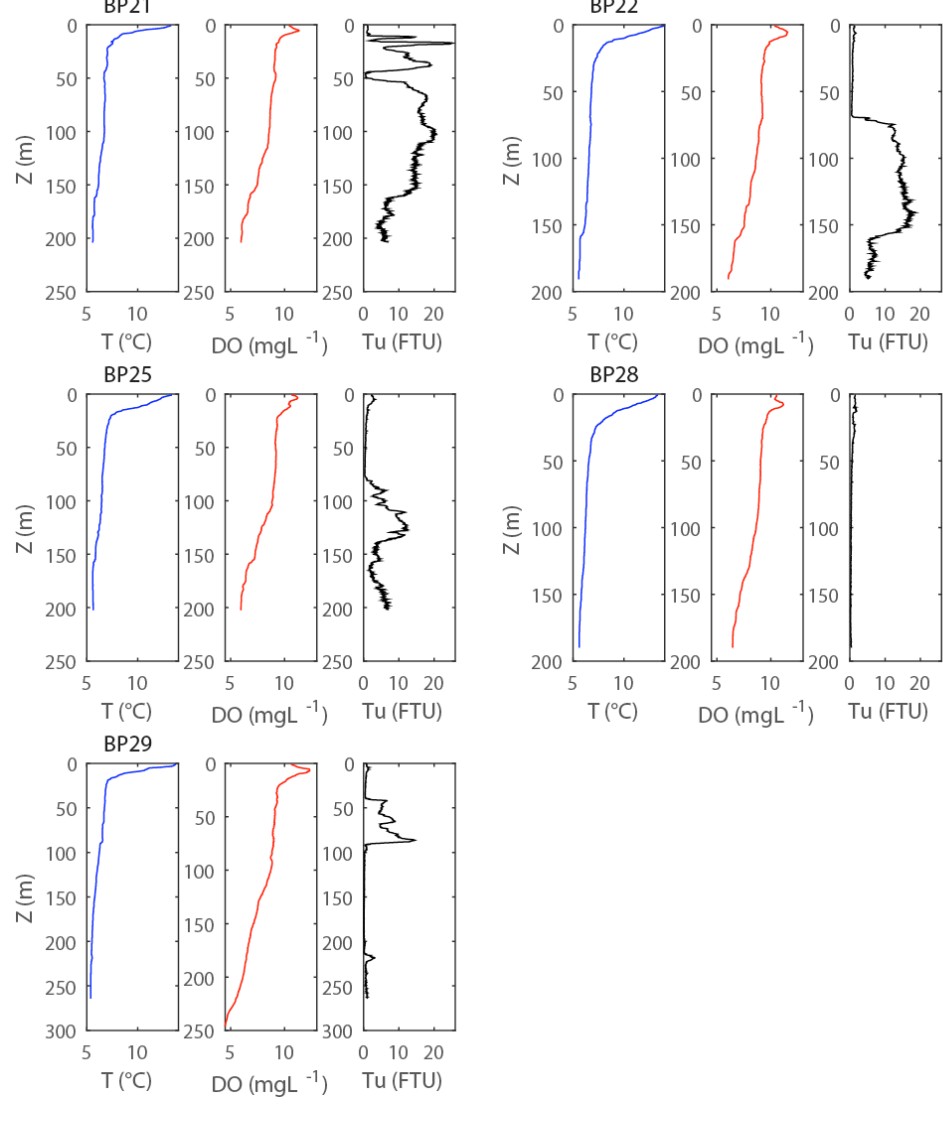





2    A2.

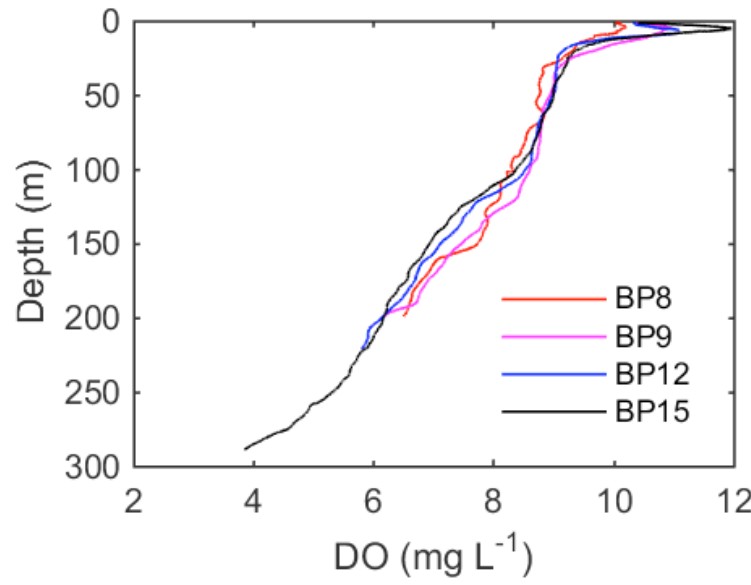



2    A3.

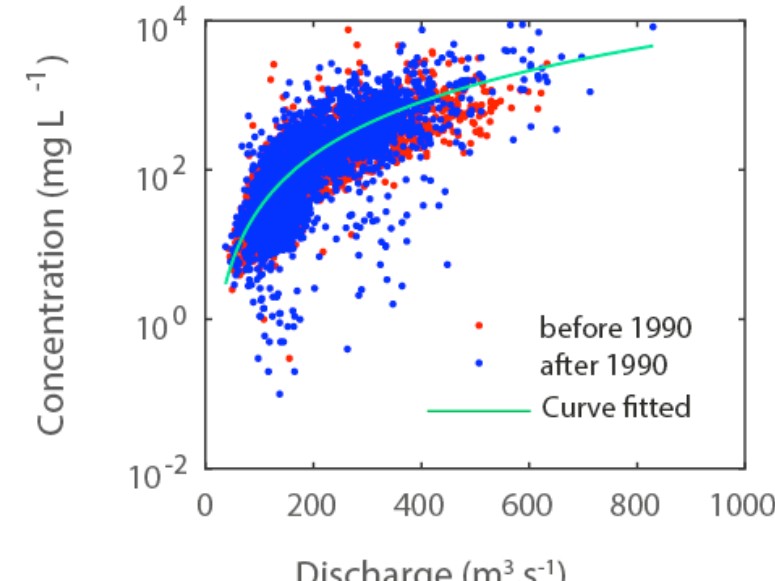

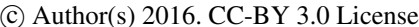



2   A4.

| BP1 | 539852 | 148149 |
|---|---|---|
| BP2 | 546865 | 146980 |
| BP3 | 546130 | 145135 |
| BP4 | 545300 | 143805 |
| BP5 | 548235 | 143210 |
| BP6 | 552530 | 141959 |
| BP7 | 550715 | 141620 |
| BP8 | 551700 | 139970 |
| BP9 | 550005 | 140470 |
| BP10 | 549005 | 139620 |
| BP11 | 547105 | 140800 |
| BP12 | 547745 | 141800 |
| BP13 | 545035 | 140800 |
| BP14 | 545565 | 142395 |
| BP15 | 542500 | 143400 |
| BP16 | 538714 | 144815 |
| BP18 | 534700 | 144950 |
| BP19 | 531138 | 142474 |
| BP21 | 530889 | 140513 |
| BP22 | 529829 | 140498 |
| BP25 | 529014 | 140688 |
| BP28 | 525772 | 144919 |
| BP29 | 532112 | 146469 |
| BP30 | 536069 | 149236 |

