# Peer review of "Are flood-driven turbidity currents hot-spots for priming"

_Biogeosciences, 2015_

## Referee Comment (RC1) · Anonymous Referee #1 · 25 Feb 2016

The MS bg-2015-645 deals with the impact of flood-driven turbidity currents on the availability of oxygen in the hypolimnion of stratified lakes. As underlined by the authors, the hypothesis of oxygen replenishments in these zones of lakes have been pointed out a long time ago but direct evidences are rare. The present study focuses on the effects of river floods on the oxygen profiles in Lake Geneva. Contrary to expectations, results presented in this study do not reveal this oxygen replenishment. On the contrary, authors observed either redistribution of oxygen into the water column without changes in mean O2 concentrations or decrease in oxygen availability in the upper part of the hypolimnion. These results are then corroborated by an experimental test (dark bioassays) showing that small river water inputs in deep lake water tend to disproportionally increase respiration, suggesting a stimulation of carbon use following river water inputs. These results are then discussed in the light of microbial co-metabolism

and priming effect mechanisms. The MS is well written, most experimental aspects are clearly described, well justified, and results are discussed in depth. I have no major comment on the MS, only questions and minor comments aimed at improving the understanding of the results.

- My first question deals with contradictory results between old observations of Meybeck et al. (1991) and those presented in the MS. Since in the present study measurements were made on a single date, could we expect that oxygen replenishment could be a transitory phenomenon? If oxygen profiles were measured throughout time, could we expect first an oxygen replenishment (in accordance with old observations) then followed by a decrease in O2 below initial values due to a stimulation of microbial respiration. For me, both results are not necessarily contradictory and this aspect could be discussed in the MS - A second question related to the first one: Even if I believe that co-metabolism and/or priming effects arise, could O2 transported in river water be a primer of lake C mineralization? This question could perhaps be partly solved - and discussed- if initial O2 levels during dark bioassays were given. - I am not specialist at all of this question but would it be interesting to discuss of O2 concentrations both in terms of saturation levels and mg L-1? The related questions are : could higher river temperature lead to saturated but "low" O2 concentrations (in mg L-1) inputs in lake water, partly explaining O2 depletions in lake water measurements? Could observations differ if river floods come from ice melt or from (warmer) spring rainfalls?

And more minor questions/comments: - P8, L5: is it really 0.22mg L-1 m-1? I am probably wrong but this value seems huge since graphically, we can see variations between ca. 10-11mg L-1 and 5-6mg L-1 between 20 m and 200m deep. Such a decrease of 0.22mg O2 L-1 would lead to O2 levels of 0 mg L-1 on a 50m deep water column. - I find the results of the dark bioassays very interesting, especially when discussed in the light of priming effect and co-metabolism. It would certainly require further testing to understand in more depth the underlying mechanisms. However, as written, I find it might be a bit confusing for readers since both mechanisms are discussed in two

distinct paragraphs. I suggest merging paragraphs 4.4 and 4.5 in a more integrated discussion. - Why the 50% treatment was not tested with lake water coming from 200m deep? - Try to justify the selected % of river water introduced in the microcosms. Is 1% still high considering the size of Lake Geneva? Or is it what could be expected during the highest floods? Obviously this might differ as a function of the position in the Lake, but such calculation could render the bioassay more convincing for explaining results observed in May. - Changes that could have occurred between Dranse water entering Lake Geneva in May and Dranse water collected for the bioassays are well discussed in the MS. However, what could have occurred to lake water during the same period? Do we expect huge changes in lake water physico-chemistry between May and bioassays especially after the important spring river floods of 2015? And really minor things. . . - P3, L1: what kind of anomalies? Positive, negative, idiosyncratic? - Why not introducing co-metabolism and priming effect concepts in the introduction? It is fundamentally not a problem, but since priming effect is used in the title of the MS, reader could expect to see this concept discussed earlier than in the last paragraph of the discussion. - P7, L26: "suspended" instead of "susepended" - P11, L21: suppress the "," after Dranse - P11, L25: Turbidity -

---

## Referee Comment (RC2) · Anonymous Referee #2 · 26 Feb 2016

This manuscript deals with the effect of flood driven turbidity currents on lake oxygen distribution and microbial respiration. It describes an extreme event in May 2015 in Lake Geneva, when heavy precipitation caused turbid river water from rivers Rhone and Dranse to push into the deeper layers of the lake. The authors have taken the opportunity to sample at selected sites in the lake that are differently affected by these turbidity currents to establish the depth-distribution of dissolved oxygen, temperature and turbidity. The context is that similar events in Lake Geneva and other lakes that are deeply stratified and experience deep water hypoxia have been hypothesized to replenish oxygen to these deep layers. However, the data to support this assumption are scarce and contradictory, and the authors set out to test this hypothesis in situ during a flooding event. They observed that turbid river intrusions redistributed dissolved oxygen in deep layers, but did not lead to a net oxygen gain. In shallower

layers, an intriguing net oxygen deficit was observed, which they interpret as a respiration over-yielding caused by priming effects, whereby riverine DOM stimulates a surplus degradation of lake DOM. In order to test this hypothesis, they performed dark respiration assays, where they mix riverine water with lake water from different depths at different mixing ratios. Similar to their field observations, they detected increased oxygen consumption when riverine water was mixed with upper hypolimnion water.

The manuscript is well-written, the context and hypotheses are relatively clearly laid out and the figures are generally illustrative and clearly presented. The original aim of the study, to test the turbidity-current deep oxygen replenishment hypothesis, appears well-founded and the results seem to provide valuable information in this regard, disproving a long-held belief and adding to the understanding of how extreme events may influence lake ecosystems. As such, the field observations alone are a valuable contribution to the literature. That being said, I am not a physical limnologist or hydrologist, and my ability to fact-check the background and interpretation of the results is therefore limited.

In principle, I am very much in favour of the approach to confirm and mechanistically test field observations by laboratory incubations, and the dark respiration assays are a clever and appropriate way to do so in this case. However, I have a number of reservations about how these incubations were carried out, and how the results from them are presented and discussed.

Firstly, the incubations were carried out in October with river water that was not turbid and likely had a very different composition of DOM and dissolved nutrients compared to during the flooding event in May. The authors themselves acknowledge this discrepancy, and argue that the aim was rather to test the responses of lake water from the different hypolimnetic layers to river water, regardless of the composition of the river water (page 13, lines 5-9). I agree that the results have some value in this regard, but they are still very unrepresentative of the context of the field observations. This makes one wonder why the respiration assays were not carried out on more occasions, at

least some of them involving flood-like conditions? A circumstance that the authors put forward, is that the October river water conveniently had the same DOM concentration as the river water, so that the dilution with lake water did not cause an overall difference in DOM (page 10, lines 1-4). However, I do not see how dilutions in the 10 to 100-fold range would cause drastic enough differences in DOM concentration to make such incubations invalid, even if the river water would have had a much higher DOM concentration compared to the lake water. It should be possible to normalize observed oxygen consumption rates to DOM concentration to obtain comparable metabolic activity measures between waters, for example. I commend the authors on submitting a manuscript, that is obviously the result of good and thorough work, less than a year after a major field campaign, and less than 6 months after experimental work. Yet, I can't help to wonder how much better the manuscript could have been if the authors would have waited until the next spring, and carried out additional respiration assays during more representative conditions. I would not let this be a ground for rejection, as there can be a number of valid reasons why such a delay in publication is not acceptable, but I recommend putting less emphasis on the incubation results, as they do not fit well in the context of flood-driven turbidity currents and they do not prove the occurrence of priming effects (see below).

Second, I am not entirely convinced that the incubation experiment in fact indicates a priming effect, since the increased oxygen consumption in the 1-10% river water in lake water mix is compared statistically to oxygen consumption in lake water alone. More appropriate in my opinion would be to compare to an expected oxygen consumption, adding the oxygen consumption of each part of the mix together. The authors do make such a comparison in the discussion, but only of a few examples are given and there is no statistical testing to support claims. See specific comments below for more detail.

Third, I can see alternative explanations than the priming effect for any disproportional increase in oxygen consumption when river water is mixed with lake water compared to when they are incubated in isolation. The authors mention nitrification (page 10, lines

12-17), and increased respiration of particulate carbon such as microbial biomass is another possibility. A budget of dissolved OM in the incubation flasks would have been a way to confirm that the observed differences in oxygen consumption were indeed a result of respiration of DOM, as the authors suggest. Yet, TOC concentrations appears to not have been measured after the incubations, or the data is not shown. Similarly, it would have been valuable to measure dissolved nutrient concentrations both before and after incubations to rule out the influence of other processes, such as nitrification or fertilization effects.

Another interesting aspect that is discussed in the manuscript is the inoculation of distinct microbial communities by the river water, or the exposure of the river DOM to distinct lake communities, that could change the OM degradation rates due to functional differences of these microbial communities (page 15, lines 16-19). This possibility could be ruled out by sterile filtration of either lake or river water prior to incubation. I am not suggesting that the authors should have done this, and they would probably have had to include a measure of microbial biomass to account for differences in respiration due to microbial biomass alone, but if they would perform similar experiments in the future it is a possibility worth considering.

All in all, it appears to me that the results of the incubation experiments performed in this study are a bit too preliminary to add much of an explanation to the field observations and to suggest that the priming effect is important in this context. The priming effect has received significant attention in aquatic ecosystems in the last 5 years, and so far the reports from different aquatic ecosystems on its importance are contradictory. The concept of the priming effect seems to be attractive to aquatic scientists, but to demonstrate priming effects experimentally is not trivial. This study adds to the body of literature that reports results suggestive of priming effects, without actually demonstrating it. Although it is a worthwhile addition to the discussion on priming effects, my opinion is that potential priming effects should not be the main message of this manuscript. Either the incubation experiments can be cut out altogether (and hopefully

be included in an exciting follow-up study where they are repeated with more rigour) or less emphasis is put on the results of these experiments, which includes changing the title and shortening the discussion. If the authors decide on this alternative, and keep the incubation experiments in the manuscript, please acknowledge the limitations of your approach more clearly in the text.

Specific comments:

Title: How about changing it to something that more closely reflects the results rather than being speculative, for example: "Flood-driven turbidity currents deplete rather than replenish oxygen in a deep stratified lake"

Page 2, line 11: correct to "dense river water"

Page 2, line 13: correct to " ... balances the force"

Page 4, line 23: What do you mean by western Europe? Many people consider Scandinavia part of western Europe and there are certainly larger lakes in both Sweden and Norway in terms of volume and depth. "western continental Europe" would be more correct.

Page 9, line 13: "... in such large system" should be either: "... such a large system" or "... such large systems".

Page 9, line 22: "Fig. 6" should be "Fig. 5".

Page 9, line 25: "lower depths" is potentially confusing, as it can be misunderstood as lower down in the lake, i.e. deeper. Use "shallower depths" instead.

Page 10, lines 22-31: I do not follow the reasoning here. How does an increase in oxygen consumption in 1-10% river water diluted in lake water compared to undiluted lake water indicate a priming effect when the river water alone has a much higher oxygen consumption compared to the lake water? You would expect an increase in oxygen consumption proportional to the amount that was added. If there is a nonadditive (disproportional) effect when waters are mixed, it should be expressed as the difference to the expected additive effect. See also comment to figure 6.

Page 26, line 26: Correct typo to "suspended"

Page 15, line 29: Correct to " ... true underflow processes"

Page 15, lines 2-10: Here the authors compare the increase in respiration to what you would expect from proportional mixing effects. These calculations should be made in the results in my opinion, and the observed respiration increase should be compared to the expected statistically for all treatments and time points. A graphical representation of these comparisons would also be a valuable addition to the figures.

Page 16, line 8: Which "underlying mechanisms" are you referring to? Could you rephrase to make this more clear?

Page 16, lines 16-17: "... river intrusions in the upper hypolimnion resulted in an increase of BOTH autochthonous and allochthonous organic matter respiration" How can you know this?

Figure 5, legend: You should indicate how many samples these boxes are based on (n=...).

Figure 6: This figure does not alone illustrate the presence of any priming effect since it is unclear how the observed increases in respiration differs from what you would expect when you mix the highly respiring river water with the relatively inactive lake water. If you use for example the end-point measurements, you would expect that an addition of 10% of river water would respire 10% of the oxygen that river water alone respires (that should be about 0.3 mg O2/l according to the y-axis values I am reading out of panel b). The 90% of lake water should respire 90% of the oxygen that it respires alone (roughly 0.7 mg O2/l) that makes 1.0 if you add them up. This is indeed lower than the ~1.5 that you observe, but is it significantly lower? I can't tell from the top of my head how you would go about to test this in a statistically sound way, but the additive effect is what

you should be comparing to, not the baseline lake respiration (as in results). Perhaps it would help to provide the expected respiration as a separate line but I fear that the plot would be too messy. You could instead choose to plot the time points separately as barcharts, with a bar representing the expected additive oxygen consumption next to the observed bar for every treatment. Alternatively you could plot every treatment separately across time in a multi-panel figure.

---

## Referee Comment (RC3) · Anonymous Referee #3 · 9 Mar 2016

General comments The MS "Are flood-driven turbidity currents hot-spots for priming effect in lakes?" proposed by Boufffard and Perga discusses a quite important process in lakes subject to partial overturn and oxygen depletion on their hypolimnion. It has long been assumed that flood-driven density current in such lakes can partly contribute to oxygen repletion of deep hypolimnion. In the present paper the authors describe the effect of a major single event in Lake Geneva that occurred in May 2015, on the oxygen concentration profile in the water column. To substantiate their hypothesis of a "priming" effect induced by the inputs of terrestrial OM, the author designed an experimental part to evaluate the respiration rate in various lake/river mixing assays. The paper presents a quite unique spatial and depth survey (in Lake Geneva) after a major event and thus these field results are very valuable. The presented evidences are relatively convincing that the effectiveness of flood-driven density currents is null or even negative in the current situation. However in the literature the role of such currents have been stressed mainly in low to very low-oxygen hypolimnion, that is not the case here. What would be the effect of the current if the hypolimnic O2 concentrations were below 2 mg/L. I recognize that the authors are cautious about their results and want to show that "turbidity currents (not) necessarily increase hypolimnetic oxygen stocks". So they should better discuss the various situations. A recurrent question is the uncertainty of the measurements. There is no mention of the reproducibility and repeatability of, for instance, the O2 measurements; therefore it is difficult for the reader to evaluate if the observed variations are significant. I agree with the two anonymous reviewers about the small representativeness of the experiment to explain what happened during the main event.

Specific comments P1 L1 "...river water supersaturated... "Generally rivers are not supersaturated with oxygen. For instance data presented for the Rhone River show a 97% saturation. So "river water saturated with oxygen" may be better. P4L24. With 309m Lake Geneva is not the deepest Western Europe lake (Lago di Garda 346m, Como 410m, Maggiore 372m). P4L25 western basin? Traditionally, Lake Geneva is divided in two main basins, the Grand-lac to the east and the petit lac to the southwest. The two main rivers empty in the Grand-lac so eastern basin. P6L28. There is a discrepancy between the text (rainfall > 100mm) and fig 1a that show > 800mm. P7 L24-27. This sentence is not clear. Why an increase in temperature suggests that the density of water is affected by suspend matter? Do the authors mean that the suspended loads compensate the temperature effect on density in order to form a density current? P8L32 This cannot be the Veveyse river, or a remote effect of it, as the Veveyse location on the map (fig 2) is wrongly positioned. The Veveyse mouth is about 7km to the southeast (6.8350°E, 46,4610°N).

Fig captions and figures Fig 2a. It is not clear what is represented on fig 2. Maximum, average turbidity?. The interpolation seems quite hypothetical and loosely constraint in some areas. The map needs a scale. Coordinates of the map are in a different system

of coordinates from the one given in appendix 4.

Technical comments P5 L19 25 sampling sites in the text, 24 on the map fig 2. Sampling site in the Eastern basin, not Western basin. P6 L5. "SHL2" site is not defined. This is the main monitoring site in Lake Geneva since decades. P9L21 Dissolved oxygen concentration units not homogenous throughout the text. Usually mg/L, and here g m-3. They are equivalent but confusing for the reader.

---

## Author Comment (AC1) · 11 Apr 2016

R1-1- Âń My first question deals with contradictory results between old observations of Meybeck et al. (1991) and those presented in the MS. Since in the present study measurements were made on a single date, could we expect that oxygen replenishment could be a transitory phenomenon? If oxygen profiles were measured throughout time, could we expect first an oxygen replenishment (in accordance with old observations) then followed by a decrease in O2 below initial values due to a stimulation of microbial respiration. For me, both results are not necessarily contradictory and this aspect could be discussed in the MS. Âż

About apparent inconsistencies with Meybeck et al's (1991) results : Meybeck et al. (1991) suggested, without yet no mean of proving it, that positive oxygen anomalies

should be associated either with winter lake cooling and subsequent along slopes oxygen-rich density current, or with river intrusion. Year 1983 shows remarkable examples of anomalies near the bottom at the deepest point of the lake (their Figure 7 reproduced below).

However the correlation with river discharge from the Rhône or the Dranse remains unclear. Deep O2 anomalies may then occur even under normal discharge condition calling for an important role of lake surface cooling in deep O2 replenishment (Figure 1). For instance, Meybeck et al (1991) observed many positive anomalies in the first three months of 1983. Yet, the weak discharges observed in January to March 1983 should not provide enough momentum to the intrusion to reach the middle of the lake and instead a quickly mixed intrusion would have been expected. It is important to notice that the intrusion reported by Meybeck et al. (1991) are found at the lake bed while our observation from the 2015 floods indicates intrusion within the water column. Their dynamics are therefore different.

In any case, such phenomena are transitory and horizontal : vertical and horizontal diffusion will rapidly smooth any positive or negative oxygen anomaly.

About an oxygen replenishment followed by a consumption: As mentioned by the reviewer, the decrease in oxygen was certainly following an increase in oxygen in the intrusion. Our measurements were carried out 3 days after the discharge peak and an oxygen sensor moored at the depth of the intrusion (for instance at BP18) will likely have recorded first a slight increase of O2 and then an excess consumption compared to another O2 sensor moored outside the intrusion layer. However a detailed analysis of this temporal evolution is not possible with our measurements and would require interesting follow up.

A revised version of the manuscript (section 4.1) will discuss in more detail the apparent discrepancy between the anomalies found in Meybeck et al (1991) and our observations based on the above mentioned points.

R1-2 Âń A second question related to the first one: Even if I believe that co-metabolism and/or priming effects arise, could O2 transported in river water be a primer of lake C mineralization? This question could perhaps be partly solved - and discussed- if initial O2 levels during dark bioassays were given. - I am not specialist at all of this question but would it be interesting to discuss of O2 concentrations both in terms of saturation levels and mg L-1? The related questions are : could higher river temperature lead to saturated but "low" O2 concentrations (in mg L-1) inputs in lake water, partly explaining O2 depletions in lake water measurements? Âż

Initial O2 concentrations were not different between incubations treatments. Overall, they ranged between 8-10 mg L-1. By the time that collected lake water was brought back to the lab and incubations started, DO had got close to equilibrium with the atmosphere. For the field data, the assumption made by the reviewer would imply that bacterial respiration/organic matter mineralization is stimulated by higher oxygen concentrations. Indeed there are some evidence of such a stimulating effect when shifting from anoxic to oxic conditions (Hulthe, G., S. Hulth, and P. O. J. Hall , 1998. Effect of oxygen on degradation rate of refractory and labile organic matter in continental margin sediments, Geochimica et Cosmochimica Acta, 62(8), 1319-1328). However, the stimulating effect has not been observed for varying oxygen concentrations within the oxic range. Whatever the considered lake depths, hypolimnetic water was always oxic, O2 being > 10 mg L-1 above 110 m and always > 4 mg L-1 even at greater depths. So priming by oxygen is unlikely to explain neither the bioassays, nor the field data. Regarding the second point, the Dranse water temperatures over the year ranges between 0 - 16°C, and more specifically 4 - 10 °C in spring. Close to oxygen saturation (see comment from rev 3, the Rhône and Dranse rivers shall be closer to 97% saturation), this would lead to O2 > 10 mg L-1 year long.

Âń Could observations differ if river floods come from ice melt or from (warmer) spring rainfalls? Âż This is likely the case since the organic matter carried by the hydrological flows shall be quite different between the two situations. A good reason to carry on

research on that topic.

R1-3 Âń And more minor questions/comments: - P8, L5: is it really 0.22mg L-1 m-1? I am probably wrong but this value seems huge since graphically, we can see variations between ca. 10-11mg L-1 and 5-6mg L-1 between 20 m and 200m deep. Such a decrease of 0.22mg O2 L-1 would lead to O2 levels of 0 mg L-1 on a 50m deep water column Âż

One decimal mistake : 0.022 mg L-1 m-1

R1-4 Âń I find the results of the dark bioassays very interesting, especially when discussed in the light of priming effect and co-metabolism. It would certainly require further testing to understand in more depth the underlying mechanisms. However, as written, I find it might be a bit confusing for readers since both mechanisms are discussed in two distinct paragraphs. I suggest merging paragraphs 4.4 and 4.5 in a more integrated discussion Âż. This would be done in an amended version of the ms

R1-5 Âż Why the 50% treatment was not tested with lake water coming from 200m deep? - Try to justify the selected % of river water introduced in the microcosms. Is 1% still high considering the size of Lake Geneva? Or is it what could be expected during the highest floods? Obviously this might differ as a function of the position in the Lake, but such calculation could render the bioassay more convincing for explaining results observed in May Âż.

Tested ranges are based on bulk estimated values of river mixing in Lake Geneva. Details are provided below. The thickness of the intrusion provides information on the dilution of the riverine water by lake water. We assume first that horizontal dispersion, $K_H$, is of same order as vertical dispersion, $K_z$ (e.g. conservative case as typically, $K_H > K_z$). The rate of dilution, $\Gamma\delta$, can be defined by $\Gamma\delta = i\delta i/\delta j$ , with $\delta$the thickness of the intrusion at the location defined by indices i and j. Taking i = BP22 or entrance of the river Dranse, respectively and j = BP18, gives $\Gamma\delta$ = 46 % (i= BP22) and $\Gamma\delta$ = 0.9 % (i = entrance of river Dranse). The rate of dilution within the intrusion can also

be estimated, assuming negligible particle settling away from the plunging point, by comparing the averaged temperature anomaly in 2 profiles (e.g. BP22 and BP18). The intrusion density, I, is a function of temperature T, and particle concentration C with I = T + C (Figure 2). I is calculated from the linearly interpolated temperature profile in the absence of intrusion. In so doing, we estimate $\Gamma$ = C, BP18/ C, BP22 = 29 % over the 4 km distance between BP22 and BP18. To have $\Gamma\delta = \Gamma$, implies to have an horizontal dispersion 1.5 times larger than the vertical dispersion and will lead to a Dranse river fraction at BP18 of 0.4%.

These bulk estimates suggest that the river water is first efficiently mixed in the underflow stage (e.g. most of the dilution is done before the intrusion reaches BP22), then, the dilution rate becomes smaller allowing the intrusion to propagate over a long distance. For SHL2 (BP18), the riverine fraction is about O(1%) and, as shown above, river contribution increases as stations are closer to the river mouths. The 50% dilution treatment was therefore quite out of the range of possible dilution, we then focussed more attention on the functional consequences of river contrubution at low fractions.

We acknowledge that specifications about why such a range of concentrations has been chosen would be an helpful information to the reader and we would add it in an amended version of the ms (in an Appendix).

R1-6 Âń Changes that could have occurred between Dranse water entering Lake Geneva in May and Dranse water collected for the bioassays are well discussed in the MS. However, what could have occurred to lake water during the same period? Do we expect huge changes in lake water physico-chemistry between May and bioassays especially after the important spring river floods of 2015? Âż

Maybe such information could be better emphasized in a revised version of the ms, but hypolimnetic water residence time is very slow in deep Lake Geneva (a matter of decades, see p14, L 7-8 ) and indeed, N, P and OC concentrations and even temperatures were highly similar between May and October 2015. These specification were

already present in the original version of the ms (L7-10, p10).

"Orthophosphate concentrations at 100 m and 200 m depth were very comparable to those recorded during the flood (13 and 29 $\mu$gP L-1 respectively at both dates) while nitrate concentrations were slightly lower (620 and 560 $\mu$gN L-1 in October, compared to 670 and 630 $\mu$gN L-1, in May 2015)."

The rest are minor comments and suggested improvements for the ms clarity that will be integrated in an amended version of the ms

―――――――――――――――

[Figure]

[Figure]

**Figure 7.** Evolution of dissolved oxygen (mg · l⁻¹) at SHL 2 on 10 consecutive profiles from January to July 1983, showing persistence of lens anomalies during a partial overturn

[Figure]

**Fig. 1.** fig7 from Meybeck et al (1991)

[Figure]

**Figure 1R1. The Dranse and Rhône rivers
discharges for a year of the Meybeck et al's
(1991) study (see also their Figure 7). Dotted
lines represent the dates at which positive
anomalies had been detected by Meybeck et
al (1991)**

**Fig. 2.** figure 1R1. The Dranse and Rhône rivers discharges for a year of the Meybeck et al's (1991) study (see also their Figure 7). Dotted lines represent the dates at which positive anomalies had been detec

[Figure]

**Figure 2R1. Density profile measured in BP18**

**Fig. 3.** figure 2R1. Density profile measured in BP18

---

## Author Comment (AC2) · 11 Apr 2016

R2-1 Firstly, the incubations were carried out in October with river water that was not turbid and likely had a very different composition of DOM and dissolved nutrients compared to during the flooding event in May. The authors themselves acknowledge this discrepancy, and argue that the aim was rather to test the responses of lake water from the different hypolimnetic layers to river water, regardless of the composition of the river water (page 13, lines 5-9). I agree that the results have some value in this regard, but they are still very unrepresentative of the context of the field observations. This makes one wonder why the respiration assays were not carried out on more occasions, at least some of them involving flood-like conditions?

Actually, in terms of nutrient and C concentrations, river water was not that much different between the May flooding event and the following October (see P10, L5-10) but we understand the reviewer's concern. This is for the exact same reason that we specified in the original manuscript that Âń this experiment did not intend to mimic conditions during the flood but instead to investigate the variability of the metabolic processes in the different hypolimnetic layers" p13, L.7-9. Ideally, the experiment should have been conducted during the studied flooding event, but as we emphasized in the introduction, based on available background, a respiration effect could hardly be anticipated. Bioassays were justified by the immediate, natural and straightforward critics we got when sending around an early version of the manuscript to colleagues for advices, i.e. the supposed-to-be refractory nature of allochthonous organic matter inputs would hamper fast and significant respiration within the lake. This is indeed the most common and shared belief in global literature on the topic. The flood we had been studied was of exceptional amplitude (a 50-yr return time at least for the Dranse river) and waiting for another year would not have anyway reproduced the field conditions. The point was then to investigate the processes underlying the observed field results, and we were lucky enough that even for different flowing conditions, bioassays results reflected very well the field conditions. This stresses out the fact these processes might not be exceptional, instead their overall contribution to the lake O2 budget gets more significant in flooding conditions. Shall we revise this manuscript, we would better emphasize that point. We agree though, and this is the next step of this ongoing work, that tests at different seasons would be quite informative.

R2-1. A circumstance that the authors put forward, is that the October river water conveniently had the same DOM concentration as the river water, so that the dilution with lake water did not cause an overall difference in DOM (page 10, lines 1-4). However, I do not see how dilutions in the 10 to 100-fold range would cause drastic enough differences in DOM concentration to make such incubations invalid, even if the river water would have had a much higher DOM concentration compared to the lake water. It should be possible to normalize observed oxygen consumption rates to DOM concentration to obtain comparable metabolic activity measures between waters, for

example.

As the reviewer might have noticed, this clearwater lake has very low DOC concentrations (about 1 mg/L). If the flooding river waters were as rich as 3-4 mg/l (which remains though a low concentration), even the 10% treatment might have changed the initial DOC concentrations by 30%-40%. In such a case, normalization would have been very helpful. In the present case, DOC concentrations were 0.8 and 0.7 mg/l for the lake water, and 0.75 mg/l for the Dranse river, which, accounting for the accuracy of the TIC-TOC equipment, might not be even significantly different. Normalization by DOC concentrations provides fully similar results. Yet, shall this representation (Fig 1 R2) give more trust to our results, it could easily substitute it to the actual figure in a revised version of the manuscript.

R2-2. I commend the authors on submitting a manuscript, that is obviously the result of good and thorough work, less than a year after a major field campaign, and less than 6 months after experimental work. Yet, I can't help to wonder how much better the manuscript could have been if the authors would have waited until the next spring, and carried out additional respiration assays during more representative conditions. I would not let this be a ground for rejection, as there can be a number of valid reasons why such a delay in publication is not acceptable, but I recommend putting less emphasis on the incubation results, as they do not fit well in the context of flood-driven turbidity currents and they do not prove the occurrence of priming effects (see below).

See reply to the previous comments. Our point of the experiment was more to test the process, that to mimic the environmental conditions of the field survey. We yet still believe these are crucial.

R2-3. Second, I am not entirely convinced that the incubation experiment in fact indicates a priming effect, since the increased oxygen consumption in the 1-10% river water in lake water mix is compared statistically to oxygen consumption in lake water alone. More appropriate in my opinion would be to compare to an expected oxygen

consumption, adding the oxygen consumption of each part of the mix together. The authors do make such a comparison in the discussion, but only of a few examples are given and there is no statistical testing to support claims. See specific comments below for more detail.

We do contest the absence of statistical testing since the O2 consumption curve over time were statistically compared between all treatments using ANCOVA (P 16, L16-24-28), such as final O2 consumption after 92h (l 21) and results are non-ambiguous. Comparing to expected O2 consumption is yet a possibility (see figure below done for un-normalized O2 consumption as DOC-normalized consumptions provide fairly similar results) and leads to the exact same conclusions : there is a substantial respiration overyield when mixing a small fraction of the Dranse water to the lake water at 100m depth. Interestingly, a 50-50 mix results in an underyield respiration. This proportion is far above what could be expected within the lake but would deserve further investigations. We believe this figure (Fig 2 R2) is redundant with the rest of the ms but could easily replace and strengthen text P 15 l 1-10.

R2-4. Third, I can see alternative explanations than the priming effect for any disproportional increase in oxygen consumption when river water is mixed with lake water compared to when they are incubated in isolation. The authors mention nitrification (page 10, lines12-17), and increased respiration of particulate carbon such as microbial biomass is another possibility. A budget of dissolved OM in the incubation flasks would have been a way to confirm that the observed differences in oxygen consumption were indeed a result of respiration of DOM, as the authors suggest. Yet, TOC concentrations appears to not have been measured after the incubations, or the data is not shown. Similarly, it would have been valuable to measure dissolved nutrient concentrations both before and after incubations to rule out the influence of other processes, such as nitrification or fertilization effects.

We agree that full demonstration for priming effect requires that the fresh organic matter is added to the incubation pre-filtered for microbial inoculum, and also that a C

mass balance (or better an artificial isotope tracking) is being done. We have been yet quite careful about these limits in this original version of the ms and these are explicitly mentioned (L16-18 p15) and discussed. The unfiltered additions are a more realistic representation of what happens in nature and C mass balance is sometimes uneasy because of the low DOC context of these waters. Increased respiration of particulate carbon would have been an explanation in the case that POC was high in the Dranse river water used for bioassays. However, for both lake and river waters, POC concentrations are basically beyond detection limits (<0.1 mg.L-1). We can still add these specifications though in a revised version of the paper. We could however reasonably rule out a fertilization effect (p10 L5-10) such as potential nitrification by restricting the data analysis to 92H. After that delay, there were obvious patterns for nitrification occurring in some vials (clear breaks in the oxygen consumption dynamics see Fig 3 R2).

R2-5 Another interesting aspect that is discussed in the manuscript is the inoculation of distinct microbial communities by the river water, or the exposure of the river DOM to distinct lake communities, that could change the OM degradation rates due to functional differences of these microbial communities (page 15, lines 16-19). This possibility could be ruled out by sterile filtration of either lake or river water prior to incubation. I am not suggesting that the authors should have done this, and they would probably have had to include a measure of microbial biomass to account for differences in respiration due to microbial biomass alone, but if they would perform similar experiments in the future it is a possibility worth considering.

We thank the reviewer for his/her wise suggestions, and these are indeed supporting ongoing research.

R2-6. All in all, it appears to me that the results of the incubation experiments performed in this study are a bit too preliminary to add much of an explanation to the field observations and to suggest that the priming effect is important in this context. The priming effect has received significant attention in aquatic ecosystems in the last

5 years, and so far the reports from different aquatic ecosystems on its importance are contradictory. The concept of the priming effect seems to be attractive to aquatic scientists, but to demonstrate priming effects experimentally is not trivial. This study adds to the body of literature that reports results suggestive of priming effects, without actually demonstrating it. Although it is a worthwhile addition to the discussion on priming effects, my opinion is that potential priming effects should not be the main message of this manuscript. Either the incubation experiments can be cut out altogether (and hopefully be included in an exciting follow-up study where they are repeated with more rigour) or less emphasis is put on the results of these experiments, which includes changing the title and shortening the discussion. If the authors decide on this alternative, and keep the incubation experiments in the manuscript, please acknowledge the limitations of your approach more clearly in the text.

We still believe that the bioassays, although not intended to mimic the full field conditions, are required to nail down flood driven respiration as a plausible process. Cutting them out would be a real weakness as most readers would only not believe in high and fast respiration of allochthonous organic matter, as this is the most shared belief in current literature. Experiments have been conducted in the light of the paper main hypothesis, i.e. they have been focussing on O2 consumption rather than C mass balance. We fully agree that these are not enough to fully demonstrate 'priming effect' and we took real care in the first version not to claim we did. Most of the limitations mentioned by the reviewer were already thoughtfully discussed in the original version but we could try to emphasize these limits a bit more in a revised version. Maybe keeping 'priming effect' in the title, although it is worded as a question and preceded by 'potential' is a bit too provocative and we would, if required, change it. But there are in this paper, strong evidence that this is a process that could take place in lake depths.

R2-7 Figure 6: This figure does not alone illustrate the presence of any priming effect since it is unclear how the observed increases in respiration differs from what you would expect when you mix the highly respiring river water with the relatively inactive

lake water. If you use for example the end-point measurements, you would expect that an addition of 10% of river water would respire 10% of the oxygen that river water alone respires (that should be about 0.3 mg O2/l according to the y-axis values I am reading out of panel b). The 90% of lake water should respire 90% of the oxygen that it respires alone (roughly 0.7 mg O2/l) that makes 1.0 if you add them up. This is indeed lower than the âĹij1.5 that you observe, but is it significantly lower? I can't tell from the top of my head how you would go about to test this in a statistically sound way, but the additive effect is what you should be comparing to, not the baseline lake respiration (as in results). Perhaps it would help to provide the expected respiration as a separate line but I fear that the plot would be too messy. You could instead choose to plot the time points separately as barcharts, with a bar representing the expected additive oxygen consumption next to the observed bar for every treatment. Alternatively you could plot every treatment separately across time in a multi-panel figure.

See figure 2 and associated reply (R2-3)

Remaining specific comments are specifications, editing or suggested improvements that will be integrated in an amended version of the ms.

———————————————

[Figure]

**Figure 1R2. Normalized Oxygen consumption (molar ratios; μmol O$_2$ per μmol of initial DOC) in the bioassays**

**Fig. 1.** Figure 1R2. Normalized Oxygen consumption (molar ratios; $\mu$mol O2 per $\mu$mol of initial DOC) in the bioassays

[Figure]

**Figure 2R2. Expected (based on a mixing model) and observed O2 consumption in the bioassays of mixed lake and river waters.**

**Fig. 2.** Fig2R2

[Figure]

Figure 7.2-1. Biochemical oxygen demand curves: (A) typical carbonaceous-demand curve showing the oxidation of organic matter, and (B) typical carbonaceous- plus nitrogenous-demand curve showing the oxidation of ammonia and nitrite. (Modified from Sawyer and McCarty, 1978.)

**Figure 3R2 a. DO concentrations over time in the example vials. The abrupt break in the slope after 80-100 h in L200-90%-t1 and L200-100% t1 are typical for nitrification processes as specified in Figure 3b.**

**Fig. 3.** Fig3R2

---

## Author Comment (AC3) · 11 Apr 2016

R3-1 In the literature the role of such currents have been stressed mainly in low to very low-oxygen hypolimnion, that is not the case here. What would be the effect of the current if the hypolimnic O2 concentrations were below 2 mg/L. I recognize that the authors are cautious about their results and want to show that "turbidity currents (not) necessarily increase hypolimnetic oxygen stocks". So they should better discuss the various situations.

Based on our dataset we cannot predict the effect of the intrusion on an anoxic hypolimnion. The interesting case of oxygen rich river water entering into anoxic water should be investigated separately as the transition from anoxic to oxic water may lead to other processes of larger importance than the priming we have hypothesized. This

limitation can be integrated in a revised version of the manuscript.

R3-2. A recurrent question is the uncertainty of the measurements. There is no mention of the reproducibility and repeatability of, for instance, the $O_2$ measurements; therefore it is difficult for the reader to evaluate if the observed variations are significant.

It is unclear whether the reviewer refers to the field or the lab data. In both case though, the measures have been performed using optical sensors that had been calibrated according to the manufacturer's guidance (Sea&Sun Technology for the field probe, Pre-Sens for the lab bioassays) just before the study. These measures are instantaneous. For the lab measures, the sensor was set on 1 measure/3s and the measures are averaged over 30s. Every treatment in the bioassays has been conducted as triplicates (L15-17, p6). Calibration of the Pre-Sens (two-point calibration DO 0-100%) was tested again at the end of the experiment (<1 week) and showed no drift over the whole duration of the bioassays.

R3-3 I agree with the two anonymous reviewers about the small representativeness of the experiment to explain what happened during the main event.

Rev 1 is actually quite supportive of the bioassays. Indeed Rev 2-3 questioned the representativeness of the experiment, but as mentioned as reply to Rev-2, the bioassays intended to test for the possibility of fast and efficient respiration of allochthonous organic matter in the hypolimnion, more than to strictly mimic the conditions during the floods. We understand the reviewer's concern and this is for the exact same reason that we claimed in the manuscript that Âń this experiment did not intend to mimic conditions during the flood but instead to investigate the variability of the metabolic processes in the different hypolimnetic layers" p13, L.7-9. Ideally, the experiment should have been conducted during the studied flooding event, but as we emphasized in the introduction, based on available background, a respiration effect could hardly be anticipated. However, without the bioassays, the first critics we got were about the supposed-tobe refractory nature of allochthonous organic matter inputs that would hamper fast and significant respiration within the lake. The flood we had been studied was of exceptional amplitude (a 50-yr return time at least for the Dranse river) and waiting for another year would not have anyway reproduced the field conditions. The point was then to investigate the processes underlying the observed field results, and we were lucky enough that even for different flowing conditions, bioassays results reflected very well the field conditions. This stresses out the fact these processes might not be exceptional, instead their overall contribution to the lake O2 budget gets more significant in flowing conditions. Shall we revise this manuscript, we would better emphasize that point.

Remaining comments are minor and would be addressed in a revised version of the manuscript.

––––––––––––––––––––––––––––––

---

## Author Response (AR1)

**Dr Marie-Elodie Perga**
**Research Scientist**
**e-mail: marie-elodie.perga@thonon.inra.fr**

Pr. Brian A. Pellerin
**Associate Editor of Biogeosciences**

**Thonon, May 19th 2016**

Dear Pr. Pellerin,

Please find enclosed a revised version of our initial manuscript "Are flood-driven turbidity currents hot-spots for priming effect in lakes?".

We performed changes in the manuscript according to our responses to reviewers stated in the biogeosciences Discussion. Briefly, the dominant critics were from reviewer #2 who asked for more powerful evidence of respiration overyield when lake and riverborne waters were mixed. As already presented in the Discussion, we added a figure and associated statistics leaving out any remaining ambiguity. We also accounted for all other comments, as specified in the following point-by-point replies and marked revised manuscript.

We thank you such as all three reviewers for their help in improving our work.
We thank you for filing this manuscript and look forward to hearing from you soon.
With best regards,

Marie-Elodie Perga

**Institut National de la Recherche Agronomique**
**Centre alpin de Recherche sur les réseaux trophiques des Ecosystèmes Limniques**
75, Avenue de Corzent F-74200 Thonon Les Bains
**Tél. :** 04 50 26 78 18– **Fax :** 04 50 26 07 60                                    **www.inra.fr**

**Are flood-driven turbidity currents hot-spots for priming effect in lakes? D. Bouffard and M-E. Perga**

**Point by point replies to reviewers**

All lines and pages numbers refer to the marked revised document.

**Replies to Reviewer #1's comments**

R1-1- 'My first question deals with contradictory results between old observations of Meybeck et al. (1991) and those presented in the MS. Since in the present study measurements were made on a single date, could we expect that oxygen replenishment could be a transitory phenomenon? If oxygen profiles were measured throughout time, could we expect first an oxygen replenishment (in accordance with old observations) then followed by a decrease in O2 below initial values due to a stimulation of microbial respiration. For me, both results are not necessarily contradictory and this aspect could be discussed in the MS. '

**About apparent inconsistencies with Meybeck et al's (1991) results :**
We added some comments in the discussion. P12, l28-32. As we did a more in-depth work in the biogeosc. Discuss. Section, and since these are published and therefore fully citeable, we also refered to Bouffard, D., Perga, M.-E. : Interactive comment on "Are flood-driven turbidity currents hot-spots for priming effect in lakes?", Biogeosciences Discuss., doi: 10.5194/bg-2015-645-AC4, 2016, for more details.

**About an oxygen replenishment followed by a consumption:**
Specifications added p17, l12-15

R1-2 ' A second question related to the first one: Even if I believe that co-metabolism and/or priming effects arise, could O2 transported in river water be a primer of lake C mineralization? This question could perhaps be partly solved - and discussed- if initial O2 levels during dark bioassays were given. - I am not specialist at all of this question but would it be interesting to discuss of O2 concentrations both in terms of saturation levels and mg L-1? The related questions are : could higher river temperature lead to saturated but "low" O2 concentrations (in mg L-1) inputs in lake water, partly explaining O2 depletions in lake water measurements? '

Specifications on initial O2 concentrations added p10 ;l21-22. A priming effect triggered by oxygen is very unlikely, as discussed in the Biogeosc. Discussion, so we did not mention it in the revised version.

- ' Could observations differ if river floods come from ice melt or from (warmer) spring rainfalls? '

This is likely the case since the organic matter carried by the hydrological flows shall be quite different between the two situations. A good reason to carry on research on that topic.

R1-3 ' And more minor questions/comments: - P8, L5: is it really 0.22mg L-1 m-1? I am probably wrong but this value seems huge since graphically, we can see variations between ca. 10-11mg L-1 and 5-6mg L-1 between 20 m and 200m deep. Such a decrease of 0.22mg O2 L-1 would lead to O2 levels of 0 mg L-1 on a 50m deep water column '

One decimal mistake : 0.022 mg L-1 m-1. This was corrected P8, l15-16.

R1-4 ' I find the results of the dark bioassays very interesting, especially when discussed in the light of priming effect and co-metabolism. It would certainly require further testing to understand in more depth the underlying mechanisms. However, as written, I find it might be a bit confusing for readers since both mechanisms are discussed in two distinct paragraphs. I suggest merging paragraphs 4.4 and 4.5 in a more integrated discussion'
We did as suggested (see p 16-17)

R1-5 ' Why the 50% treatment was not tested with lake water coming from 200m deep? - Try to justify the selected % of river water introduced in the microcosms. Is 1% still high considering the size of Lake Geneva? Or is it what could be expected during the highest floods? Obviously this might differ as a function of the position in the Lake, but such calculation could render the bioassay more convincing for explaining results observed in May '.

Tested ranges are based on bulk estimated values of river mixing in Lake Geneva. Details are provided in appendix A1, and reference is given P6, l15-17.

R1-6 'Changes that could have occurred between Dranse water entering Lake Geneva in May and Dranse water collected for the bioassays are well discussed in the MS. However, what could have occurred to lake water during the same period? Do we expect huge changes in lake water physico-chemistry between May and bioassays especially after the important spring river floods of 2015'

We did our best to emphasize such information all over the revised version of the ms (p14, l1-13), although many were already given.(p10 l 15-20)

Following minor comments were either redundant with previous corrections or editing. Everything was accounted for.

**Replies to Reviewer #2's comments**

Firstly, the incubations were carried out in October with river water that was not turbid and likely had a very different composition of DOM and dissolved nutrients compared to during the flooding event in May. The authors themselves acknowledge this discrepancy, and argue that the aim was rather to test the responses of lake water from the different hypolimnetic layers to river water, regardless of the composition of the river water (page 13, lines 5-9). I agree that the results have some value in this regard, but they are still very unrepresentative of the context of the field observations. This makes one wonder why the respiration assays were not carried out on more occasions, at least some of them involving flood-like conditions?

Consistently with our statements in the biogeosc. Discussion, we feel these bioassays, although not conducted in the same circumstances, are absolutely necessary to illustrate that these mechanisms are possible. We added further justifications within the revised version, always trying to clearly state that these did not intend to strickly mimic the flood situation. P14, l1-13. P17, l27-29.

A circumstance that the authors putforward, is that the October river water conveniently had the same DOM concentration as the river water, so that the dilution with lake water did not cause an overall difference in DOM (page 10, lines 1-4). However, I do not see how dilutions in the 10 to 100-fold range would cause drastic enough differences in DOM concentration to make such incubations invalid, even if the river water would have had a much higher DOM

concentration compared to the lake water. It should be possible to normalize observed oxygen consumption rates to DOM concentration to obtain comparable metabolic activity measures between waters, for example.

As mentioned in the BioGeosc. Discussion, normalizing oxygen consumption rates to DOM does not affect the results since DOC concentrations are always the same between lake and riverborne waters. To counteract such a concern, we added this specification in the revised version P10, I11-14.

I commend the authors on submitting a manuscript, that is obviously the result of good and thorough work, less than a year after a major field campaign, and less than 6 months after experimental work. Yet, I can't help to wonder how much better the manuscript could have been if the authors would
have waited until the next spring, and carried out additional respiration assays during more representative conditions. I would not let this be a ground for rejection, as there can be a number of valid reasons why such a delay in publication is not acceptable, but I recommend putting less emphasis on the incubation results, as they do not fit well in the context of flood-driven turbidity currents and they do not prove the occurrence of priming effects (see below).

See reply to the first comment. Our point of the experiment was more to test the process, that to mimic the environmental conditions of the field survey. We yet still believe these are crucial.

Second, I am not entirely convinced that the incubation experiment in fact indicates a priming effect, since the increased oxygen consumption in the 1-10% river water in lake water mix is compared statistically to oxygen consumption in lake water alone. More appropriate in my opinion would be to compare to an expected oxygen consumption, adding the oxygen consumption of each part of the mix together. The authors do make such a comparison in the discussion, but only of a few examples are given and there is no statistical testing to support claims. See specific comments below for more detail.

We contested that there was no statistical testing in the previous version, but as suggested by the reviewer, we added a comparison between observed and expected O2 consumption (Fig 6c- P11, I7-9, P11, I14-16) with further statistical testing. We also standardized all results to a 86h experimental duration to clarify the results.

Third, I can see alternative explanations than the priming effect for any disproportional increase in oxygen consumption when river water is mixed with lake water compared to when they are incubated in isolation. The authors mention nitrification (page 10, lines12-17), and increased respiration of particulate carbon such as microbial biomass is another possibility. A budget of dissolved OM in the incubation flasks would have been a way to confirm that the observed differences in oxygen consumption were indeed a result of respiration of DOM, as the authors suggest. Yet, TOC concentrations appears to not have been measured after the incubations, or the data is not shown. Similarly, it would have been valuable to measure dissolved nutrient concentrations both before and after incubations to rule out the influence of other processes, such as nitrification or fertilization effects.

As already discussed in the biogeos. Discussion, nitrification and fertilization could be reasonably ruled out, (such as respiration of particulate organic carbon) and we added specifications on that point. P17, I27-29.

Another interesting aspect that is discussed in the manuscript is the inoculation of distinct microbial communities by the river water, or the exposure of the river DOM to distinct lake communities, that could change the OM degradation rates due to functional differences of these microbial communities (page 15, lines 16-19). This possibility could be ruled out by sterile filtration of either lake or river water prior to incubation.
I am not suggesting that the authors should have done this, and they would probably have had to include a measure of microbial biomass to account for differences in respirationdue to microbial biomass alone, but if they would perform similar experiments in the future it is a possibility worth considering.

We thank the reviewer for his/her wise suggestions, and these are indeed supporting on going research.

All in all, it appears to me that the results of the incubation experiments performed in this study are a bit too preliminary to add much of an explanation to the field observations and to suggest that the priming effect is important in this context. The priming effect has received significant attention in aquatic ecosystems in the last 5 years, and so far the reports from different aquatic ecosystems on its importance are contradictory. The concept of the priming effect seems to be attractive to aquatic scientists, but to demonstrate priming effects experimentally is not trivial. This study adds to the body
of literature that reports results suggestive of priming effects, without actually demonstrating it. Although it is a worthwhile addition to the discussion on priming effects, my opinion is that potential priming effects should not be the main message of this manuscript. Either the incubation experiments can be cut out altogether (and hopefully be included in an exciting follow-up study where they are repeated with more rigour) or less emphasis is put on the results of these experiments, which includes changing the title and shortening the discussion. If the authors decide on this alternative, and
keep the incubation experiments in the manuscript, please acknowledge the limitations of your approach more clearly in the text.

We still believe that the bioassays, although not strictly mimicking the full field conditions, are required to mail down flood driven respiration as a plausible process. Cutting them out would be a real weakness as most readers would only not believe in high and fast respiration of allochthonous organic matter, as this is the most shared belief in current literature. Experiments have been conducted in light of the paper main hypothesis, so have been focussing on $O_2$ consumption rather than C mass balance. We fully agree that these are not enough to fully demonstrate 'priming effect' and we took real care in the first version and in the revised version not to claim we did. P18, I13-14.

**Specific comments:**

| Comments | Corrections |
|---|---|
| Title: How about changing it to something that more closely reflects the results rather than being speculative, for example: "Flood-driven turbidity currents deplete rather than replenish oxygen in a deep stratified lake" | We kept the title as it was |
| Page 15, lines 2-10: Here the authors compare the increase in respiration to what you would expect from proportional mixing effects. These calculations should be made in the results in my opinion, and the observed respiration increase should be compared to the expected statistically for all treatments and time points. A graphical representation of these comparisons would also be a valuable addition to the figures. | Done, see figure 6c and P11 |
| Page 10, lines 22-31: I do not follow the reasoning here. How does an increase in oxygen consumption in 1-10% river water diluted in lake water compared to undiluted lake water indicate a priming effect when the river water alone has a much higher oxygen consumption compared to the lake | Done, see figure 6c and P11 |

| | |
|---|---|
| water? You would expect an increase in oxygen consumption proportional to the amount that was added. If there is a non-additive (disproportional) effect when waters are mixed, it should be expressed as the difference to the expected additive effect. See also comment to figure 6. | |
| Page 15, lines 2-10: Here the authors compare the increase in respiration to what you would expect from proportional mixing effects. These calculations should be made in the results in my opinion, and the observed respiration increase should be compared to the expected statistically for all treatments and time points. A graphical representation of these comparisons would also be a valuable addition to the figures. | Same comment. We performed the mixing model to evaluate expected respiration as compared to observed. See figure 6c and P11 |
| Page 16, line 8: Which "underlying mechanisms" are you referring to? Could you rephrase to make this more clear? | Corrected to clear it up. P17 l19-21 |
| Figure 5, legend: You should indicate how many samples these boxes are based on (n=...). | Done, see fig 5 caption |
| Figure 6: This figure does not alone illustrate the presence of any priming effect since it is unclear how the observed increases in respiration differs from what you would expect when you mix the highly respiring river water with the relatively inactive lake water. If you use for example the end-point measurements, you would expect that an addition of 10% of river water would respire 10% of the oxygen that river water alone respires (that should be about 0.3 mg O2/l according to the y-axis values I am reading out of panel b). The 90% of lake water should respire 90% of the oxygen that it respires alone (roughly 0.7 mg O2/l) that makes 1.0 if you add them up. This is indeed lower than the ~1.5 that you observe, but is it significantly lower? I can't tell from the top of my head how you would go about to test this in a statistically sound way, but the additive effect is what you should be comparing to, not the baseline lake respiration (as in results). Perhaps it would help to provide the expected respiration as a separate line but I fear that the plot would be too messy. You could instead choose to plot the time points separately as barcharts, with a bar representing the expected additive oxygen consumption next to the observed bar for every treatment. Alternatively you could plot every treatment separately across time in a multi-panel figure. | Same comment again, see Fig 6c
We did not plot every treatment in a different panel but added a figure. |

**Replies to Reviewer #3's comments**

In the literature the role of such currents have been stressed mainly in low to very low-oxygen hypolimnion, that is not the case here. What would be the effect of the current if the hypolimnic O2 concentrations were below 2 mg/L. I recognize that the authors are cautious about their results and want to show that "turbidity currents (not) necessarily increase hypolimnetic oxygen stocks". So they should better discuss the various situations. A recurrent question is the uncertainty of the measurements. There is no mention of the reproducibility and repeatability of, for instance, the O2 measurements; therefore it is difficult for the reader to evaluate if the observed variations are significant.

Specifications added P6, l23-24.

I agree with the two anonymous reviewers about the small representativeness of the experiment to explain what happened during the main event.

Rev 1 is actually quite supportive of the bioassays. Indeed Rev 2-3 questioned the representativeness of the experiment, but as mentioned as reply to Rev-2, the bioassays intend to test for the possibility of fast and efficient respiration of allochthonous organic matter in the hypolimnion, more than to strictly mimic the conditions during the floods. We understand the reviewer's concern and this is for the exact same reason that we claimed in the manuscript that « this experiment did not intend to mimic conditions during the flood but instead to investigate the variability of the metabolic processes in the different hypolimnetic layers"

We emphasized that point all over the ms. P14, l1-13.

**Specific comments**

| Comments | Corrections |
|---|---|
| Specific comments P1 L1 "...river water supersaturated... "Generally rivers are not supersaturated with oxygen. For instance data presented for the Rhone River show a 97% saturation. So "river water saturated with oxygen" may be better. | Done, P1, l14 |
| P4L24. With 309m Lake Geneva is not the deepest Western Europe lake (Lago di Garda 346m, Como 410m, Maggiore 372m). P4L25 western basin? Traditionally, Lake Geneva is divided in two main basins, the Grand-lac to the east and the petit lac to the southwest. The two main rivers empty in the Grand-lac so eastern basin. | Corrected. P4, l23-26 |
| P6L28. There is a discrepancy between the text (rainfall > 100mm) and fig 1a that show > 800mm. | True, Fig 1, corrected |
| P7 L24-27. This sentence is not clear. Why an increase in temperature suggests that the density of water is affected by suspend matter? Do the authors mean that the suspended loads compensate the temperature effect on density in order to form a density current? | The sentence was amended to make this clearer P8, l2-4 |
| P8L32 This cannot be the Veveyse river, or a remote effect of it, as the Veveyse location on the map (fig 2) is wrongly positioned. The Veveyse mouth is about 7km to the southeast (6.8350◦E, 46,4610◦N). Fig captions and figures Fig 2a. It is not clear what is represented on fig 2. Maximum, average turbidity?. The interpolation seems quite hypothetical and loosely constraint in some areas. The map needs a scale. Coordinates of the map are in a different system of coordinates from the one given in appendix 4 | Map corrected (fig 2) And specifications added to the caption Coordinates in appendix (which is now 5) corrected |
| Technical comments P5 L19 25 sampling sites in the text, 24 on the map | Corrected in the text p5 |
| P9L21 Dissolved oxy- gen concentration units not homogenous throughout the text. Usually mg/L, and here g m-3. They are equivalent but confusing for the reader. | Corrected throughout the text (p9-10) |
| fig 2. Sampling site in the Eastern basin, not Western basin | Corrected p5, l19 |

[revised manuscript text omitted]

Figure 1

[Figure]

[Figure]

[Figure]

[Figure]

Figure 2

[Figure]

[Figure]

Figure 3

[Figure]

Figure 4

[Figure]

[Figure]

Figure 5

[Figure]

[Figure]

Figure 6

**Appendix**

A1. The thickness of the intrusion provides information on the dilution of the riverine water by lake water. We assume first that horizontal dispersion, $K_H$, is of same order as vertical dispersion, $K_z$ (e.g. conservative case as typically, $K_H > K_z$). The rate of dilution, $\Gamma_\delta$, can be defined by $\Gamma_\delta = \delta i / \delta j$, with $\delta$ the thickness of the intrusion at the location defined by indices i and j. Taking i = BP22 or entrance of the river Dranse, respectively and j = BP18, gives $\Gamma_\delta$ = 46 % (i= BP22) and $\Gamma_\delta$ = 0.9 % (i = entrance of river Dranse).

The rate of dilution within the intrusion can also be estimated, assuming negligible particle settling away from the plunging point, by comparing the averaged temperature anomaly in 2 profiles (e.g. BP22 and BP18). The intrusion density, $\rho_I$, is a function of temperature $\rho_T$, and particle concentration $\rho_C$ with $\rho_I = \rho_T + \rho_C$ (Figure 2). $\rho_I$ is calculated from the linearly interpolated temperature profile in the absence of intrusion. In so doing, we estimate $\Gamma_\rho = \rho_{C, BP18} / \rho_{C, BP22}$ = 29 % over the 4 km distance between BP22 and BP18. To have $\Gamma_\delta = \Gamma_\rho$, implies to have an horizontal dispersion 1.5 times larger than the vertical dispersion and will lead to a Dranse river fraction at BP18 of 0.4%. These bulk estimates suggest that the river water is first efficiently mixed in the underflow stage (e.g. most of the dilution is done before the intrusion reaches BP22), then, the dilution rate becomes smaller allowing the intrusion to propagate over a long distance.

For SHL2 (BP18), the riverine fraction is about O(1%) and, as shown above, river contribution increases as stations are closer to the river mouths. The 50% dilution treatment was therefore quite out of the range of possible dilution, we then focussed more attention on the functional consequences of river contribution at low fractions.A1.

[Figure]

[Figure]

[Figure]

[Figure]

A2.

[Figure]

[Figure]

[Figure]

[Figure]

[Figure]

A3.

[Figure]

[Figure]

A4.

[Figure]

| BP1 | 6.65535539852 | 46.48196148149 |
| BP2 | 6.74681546865 | 46.47204146980 |
| BP3 | 6.73746546130 | 46.45538145135 |
| BP4 | 6.72681545300 | 46.44335143805 |
| BP5 | 6.76507548235 | 46.43823143210 |
| BP6 | 6.82108552530 | 46.42729141959 |
| BP7 | 6.79751550715 | 46.42411141620 |
| BP8 | 6.81049551700 | 46.40934139970 |
| BP9 | 6.78839550005 | 46.41372140470 |
| BP10 | 4.77548549005 | 46.40600139620 |
| BP11 | 6.75064547105 | 46.41646140800 |
| BP12 | 6.75885547745 | 46.42551141800 |
| BP13 | 6.72372545035 | 46.41630140800 |
| BP14 | 6.73042545565 | 46.43069142395 |
| BP15 | 6.69043542500 | 46.43947143400 |
| BP16 | 6.64098538714 | 46.45186144815 |
| BP18 | 6.58872534700 | 46.45270144950 |
| BP19 | 6.54273531138 | 46.43007142474 |
| BP21 | 6.53978530889 | 46.41241140513 |
| BP22 | 6.52600529829 | 46.41216140498 |
| BP25 | 6.51537529014 | 46.41379140688 |
| BP28 | 6.47253525772 | 46.45149144919 |
| BP29 | 6.55481532112 | 46.46611146469 |
| BP30 | 6.60594536069 | 46.49139149236 |